# Diverse modes of synaptic signaling, regulation, and plasticity distinguish two classes of *C. elegans* glutamatergic neurons

Donovan Ventimiglia, Cornelia I Bargmann*

Lulu and Anthony Wang Laboratory of Neural Circuits and Behavior, The Rockefeller University, New York, United States

**Abstract** Synaptic vesicle release properties vary between neuronal cell types, but in most cases the molecular basis of this heterogeneity is unknown. Here, we compare in vivo synaptic properties of two neuronal classes in the *C. elegans* central nervous system, using VGLUT-pHluorin to monitor synaptic vesicle exocytosis and retrieval in intact animals. We show that the glutamatergic sensory neurons AWC[ON] and ASH have distinct synaptic dynamics associated with tonic and phasic synaptic properties, respectively. Exocytosis in ASH and AWC[ON] is differentially affected by SNARE-complex regulators that are present in both neurons: phasic ASH release is strongly dependent on UNC-13, whereas tonic AWC[ON] release relies upon UNC-18 and on the protein kinase C homolog PKC-1. Strong stimuli that elicit high calcium levels increase exocytosis and retrieval rates in AWC[ON], generating distinct tonic and evoked synaptic modes. These results highlight the differential deployment of shared presynaptic proteins in neuronal cell type-specific functions.
DOI: https://doi.org/10.7554/eLife.31234.001

*For correspondence:
cori@rockefeller.edu

Competing interests: The authors declare that no competing interests exist.

## Introduction

Neurotransmitter release is a highly regulated process that varies at different synapses, and at the same synapse over time (*Atwood and Karunanithi, 2002*). Although presynaptic diversity is widely observed, it is challenging to define its role in intact circuits under physiological patterns of activity (*Regehr, 2012*); more often, a synapse is examined ex vivo at calcium concentrations or temperatures that alter its properties. Studies in the zebrafish retina represent one example in which synapses of two distinct neuronal classes, ON- and OFF-bipolar cells, have been compared in vivo in intact animals, leading to insights into their similarities and differences (*Odermatt et al., 2012*). Here, we extend this approach to the central nervous system of the nematode worm *Caenorhabditis elegans*, and relate diversity in synaptic properties to requirements for specific synaptic proteins in individual neurons.

The well-studied neural circuitry of *C. elegans*, which includes 302 neurons and about 9000 synaptic connections, presents an opportunity to study presynaptic diversity in a well-defined context (*Varshney et al., 2011*; *White et al., 1986*). Most synaptic proteins are conserved between *C. elegans* and other animals; indeed, behavioral genetics in *C. elegans* led to the initial identification of the SNARE (soluble N-ethylmaleimide–sensitive factor attachment receptor) regulatory proteins *unc-13* and *unc-18* (*Brenner, 1974*; *Gengyo-Ando et al., 1993*; *Maruyama and Brenner, 1991*). However, the study of synaptic transmission in *C. elegans* has been largely limited to the neuromuscular junction (NMJ) due to the challenges of electrophysiology in this small animal (*Richmond and Jorgensen, 1999*). As a result, the synaptic properties of neurons in the central nervous system have only begun to be explored (*Lindsay et al., 2011*).

Several reporters of synaptic activity that are suitable for in vivo analysis are based on pHluorin, a highly pH-sensitive variant of the green fluorescent protein (*Miesenböck et al., 1998*). pHluorin and its derivatives are minimally fluorescent at the acidic pH conditions characteristic of the synaptic vesicle lumen, but highly fluorescence at neutral extracellular pH. As a result, synaptic vesicle exocytosis results in a sharp increase in the fluorescence of pHluorin fusion proteins targeted to the synaptic vesicle lumen. Their subsequent endocytosis and reacidification quenches fluorescence, providing readouts at multiple stages of the synaptic vesicle cycle (*Di Giovanni and Sheng, 2015*; *Fernandez-Alfonso and Ryan, 2008*; *Li et al., 2005*; *Sankaranarayanan and Ryan, 2000*).

The genetic tractability and transparency of *C. elegans* are ideal for pHluorin imaging, and indeed, pHlourin-synaptobrevin fusion proteins have been used to study steady-state synaptic properties at the neuromuscular junction and in several other neurons (*Dittman and Kaplan, 2006*; *Oda et al., 2011*; *Voglis and Tavernarakis, 2008*). However, the standing plasma membrane levels of pHlourin-synaptobrevin fusion proteins make them ill-suited to real-time analysis (*Dittman and Kaplan, 2006*). By contrast, the vesicular glutamate transporter (VGLUT) has a minimal residence time on the plasma membrane in mammalian neurons (*Voglmaier et al., 2006*), and therefore is better suited for pHluorin imaging of vesicle dynamics (*Balaji and Ryan, 2007*).

We show here that EAT-4 VGLUT-pHluorin fusions can be used to study dynamic release and retrieval of synaptic vesicles from individual neurons in intact *C. elegans*. Using VGLUT-pHluorin fusions, we show that the release and retrieval of glutamatergic synaptic vesicles in two sensory neurons, AWC[ON] and ASH, are kinetically distinct and matched to their signaling properties. We further demonstrate differential contributions of SNARE regulators to synaptic dynamics in AWC[ON] and ASH, and describe activity-dependent regulation of AWC[ON] exo- and endocytosis.

## Results

### VGLUT-pHluorin reports synaptic activity in AWC[ON] and ASH neurons

The AWC[ON] and ASH sensory neurons, which sense attractive odors and repulsive chemical and physical stimuli, respectively, are dynamically and molecularly distinct (*Bargmann, 2006*; *Serrano-Saiz et al., 2013*). Based on calcium imaging studies, the AWC[ON] olfactory neurons are tonically active at rest, inhibited (likely hyperpolarized) by odor stimuli, and transiently activated upon odor removal before a return to baseline (*Chalasani et al., 2007*; *Gordus et al., 2015*; *Kato et al., 2014*) (*Figure 1A*). This pattern resembles that of vertebrate photoreceptors, which employ a similar cGMP sensory transduction cascade (*Bargmann, 2006*; *Zhang and Cote, 2005*). By contrast, calcium imaging and electrophysiological studies of the ASH nociceptive neurons indicate that they are strongly activated by noxious chemical and mechanical stimuli, and recover quickly upon stimulus removal (*Chatzigeorgiou et al., 2013*; *Hilliard et al., 2005*; *Kato et al., 2014*)(*Figure 1B*). Like vertebrate nociceptive neurons, ASH neurons signal via G protein-regulated TRPV channels (*Hilliard et al., 2005*; *Kato et al., 2014*). Both AWC[ON] and ASH signal to downstream neurons through glutamatergic synapses and the vesicular glutamate transporter EAT-4 (*Chalasani et al., 2007*; *Lee et al., 1999*).

To image synaptic vesicle (SV) endo- and exocytosis from single neurons in intact animals, we inserted super-ecliptic pHluorin into the first lumenal domain of EAT-4 (VGLUT-pH) and expressed this fusion protein using cell-specific promoters for AWC[ON] and ASH (*Figure 1—figure supplement 1A–C*). Immobilized animals were imaged in microfluidic chips that enable the precise delivery and removal of chemical stimuli and simultaneous monitoring of cell fluorescence at high magnification (*Chalasani et al., 2007*; *Chronis et al., 2007*). AWC[ON] responses were elicited by addition and removal of butanone odor, while ASH responses were elicited by addition and removal of a noxious 0.5 M NaCl stimulus, under conditions that gave robust signals with GCaMP calcium sensors (*Figure 1A,B*).

The VGLUT-pH reporter was exclusively localized to the axon in both AWC[ON] and ASH, with a semi-punctate distribution in synaptic regions (*Figure 1C*, left panels). Delivery of butanone to AWC[ON] resulted in a reduction of VGLUT-pH fluorescence with recovery after odor removal (*Figure 1C*, top). Delivery of NaCl to ASH resulted in an increase in VGLUT-pH fluorescence, followed by a decrease after NaCl removal (*Figure 1C*, bottom). In both neurons, responses were observed across the ~20 μm region of the axon that was imaged, and could be followed with single

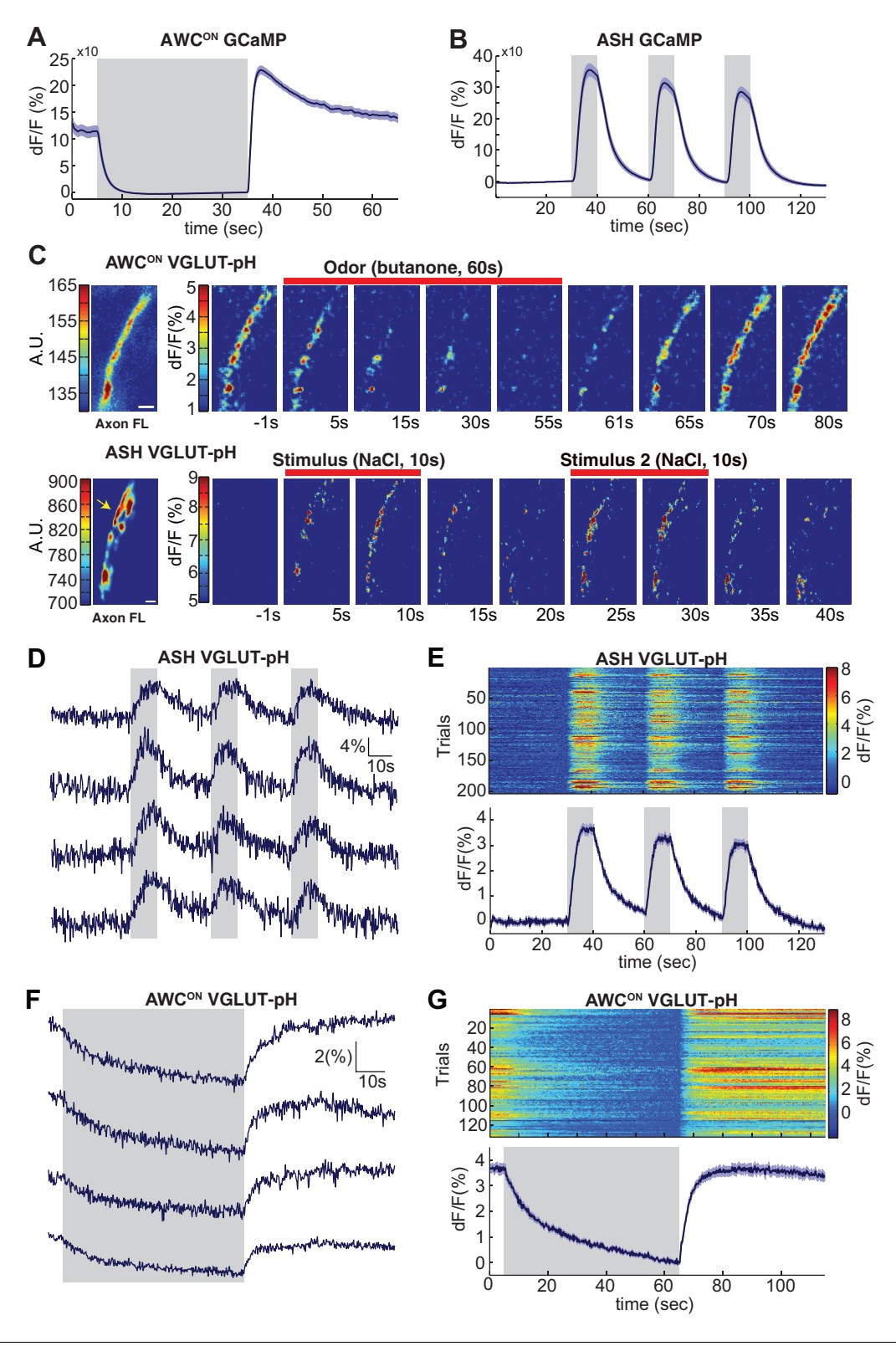

**Figure 1.** Sensory stimuli evoke VGLUT-pH signals in AWC$^{ON}$ and ASH neurons. (**A**) AWC$^{ON}$ GCaMP5A responses in the cell body upon butanone stimulation (n = 47, 16 animals, 2–3 trials each). (**B**) ASH GCaMP3 responses in the cell body upon 500 mM NaCl stimulation (n = 39, 13 animals, three trials each). Gray shaded areas mark stimulus period. (**C**) Individual AWC$^{ON}$ and ASH VGLUT-pH responses. Left: Fluorescence intensity of VGLUT-pH along the

*Figure 1 continued on next page*

*Figure 1 continued*

axon prior to stimulation (a.u. arbitrary units). White scale bars = 2 um. Right: Images of VGLUT-pH fluorescence changes upon butanone (AWC$^{ON}$) or NaCl (ASH) stimulation at t = 0 (red bars), presented as change in fluorescence versus reference F (F at ~t = 55 s for AWC$^{ON}$, t=-1s for ASH, see Materials and methods). Recordings were smoothed using a running average (three frames, three pixels in x and y). (D,F) Single trials of ASH (D) and AWC$^{ON}$ (F) VGLUT-pH responses from four individuals. Top trace in each panel is from the axon in (C). (E,G) Population ASH (E) and AWC$^{ON}$ (G) VGLUT-pH responses. Top panel: Heat map of individual trials, three trial per animal, presented in sequential order. Bottom panel: Mean response from all trials. AWC$^{ON}$: 132 trials from 44 animals, three trials each. ASH: 204 trials from 68 animals, three trials each. Blue shading around traces indicates S.E.M.

DOI: https://doi.org/10.7554/eLife.31234.002

The following source data and figure supplements are available for figure 1:

**Source data 1.** Source data for *Figure 1*.

DOI: https://doi.org/10.7554/eLife.31234.005

**Figure supplement 1.** Anatomy of AWC$^{ON}$ and ASH and illustration of synaptic reporters.

DOI: https://doi.org/10.7554/eLife.31234.003

**Figure supplement 2.** VGLUT-pH data acquisition.

DOI: https://doi.org/10.7554/eLife.31234.004

trial resolution (*Figure 1D–G* & Materials and methods). AWC$^{ON}$ and ASH synapses appeared to be highly reliable, as over 97% of stimuli triggered a detectable VGLUT-pH response (*Figure 1E,G*).

In ASH, VGLUT-pH fluorescence rose rapidly upon stimulus addition, and fell immediately upon stimulus removal, closely resembling the calcium response (*Figure 1D,E*, compare 1B). These results are consistent with a model in which ASH synaptic vesicle exocytosis is induced by stimulus-triggered calcium entry, and terminates rapidly with subsequent endocytosis and reacidification. In a control experiment, pHluorin tethered to the extracellular face of the ASH plasma membrane as a CD4 fusion protein did not respond to NaCl with fluorescence changes (*Figure 1—figure supplement 1D–E*), indicating that the signal reflects synaptic vesicle dynamics and not changes in extracellular pH.

In AWC$^{ON}$, VGLUT-pH fluorescence decreased throughout a one minute odor presentation, and odor removal resulted in a rapid increase to pre-stimulus levels without an overshoot (*Figure 1F,G*). The properties are consistent with a kinetic model in which the AWC$^{ON}$ neuron has tonic synaptic vesicle release, with basal VGLUT-pH fluorescence determined by steady-state levels of exocytosis versus endocytosis and reacidification. In this model, odor addition reduces calcium and synaptic vesicle exocytosis, and odor removal triggers a calcium increase, synaptic vesicle exocytosis, and a return to the pre-stimulus steady state.

A close examination of VGLUT-pH signals after odor removal showed that AWC$^{ON}$ fluorescence had an initial fast rate of increase for ~1 s, and then transitioned to a slower rate of increase over the following ~5 s (*Figure 2A,B*, *Figure 2—figure supplement 1A*). These dynamics suggest that synaptic vesicle exocytosis is transiently enhanced above its basal level immediately after odor removal, in agreement with the known calcium overshoot in the AWC$^{ON}$ cell body (e.g. *Figure 1A*). To better understand the relationship between somatic calcium, synaptic calcium, and exocytosis in AWC$^{ON}$, we fused a GCaMP calcium sensor to the synaptic vesicle protein synaptogyrin (*Figure 1—figure supplement 1C*). This protein labeled the axons in a punctate pattern consistent with synapses, and should detect calcium levels in the immediate vicinity of synaptic vesicles. Synaptic calcium levels monitored with synaptogyrin-GCaMP rose and fell substantially more quickly than those in the cell body (*Figure 2C,D*; compare *Figure 1A*). The peak rate of calcium entry, at ~1 s after odor removal, occurred at the same time as the peak rate of VGLUT-pH fluorescence increase (*Figure 2B,D,G*).

Varying the duration of odor exposure prior to odor removal allowed a more focused comparison of synaptic calcium dynamics and VGLUT-pH dynamics in AWC$^{ON}$ (*Figure 2C–F*). Removing odor after a 10 s exposure elicited a small synaptic calcium overshoot within the first second (*Figure 2D, G*), and a small increase in exocytosis during the same interval (*Figure 2H*). Both synaptic calcium and exocytosis rates were elevated more substantially for ~1 s after a 60 s or 180 s odor exposure (*Figure 2F–H*). This correspondence suggests that the transient synaptic calcium overshoot following long odor stimuli evokes a brief pulse of synaptic vesicle release above the tonic level.

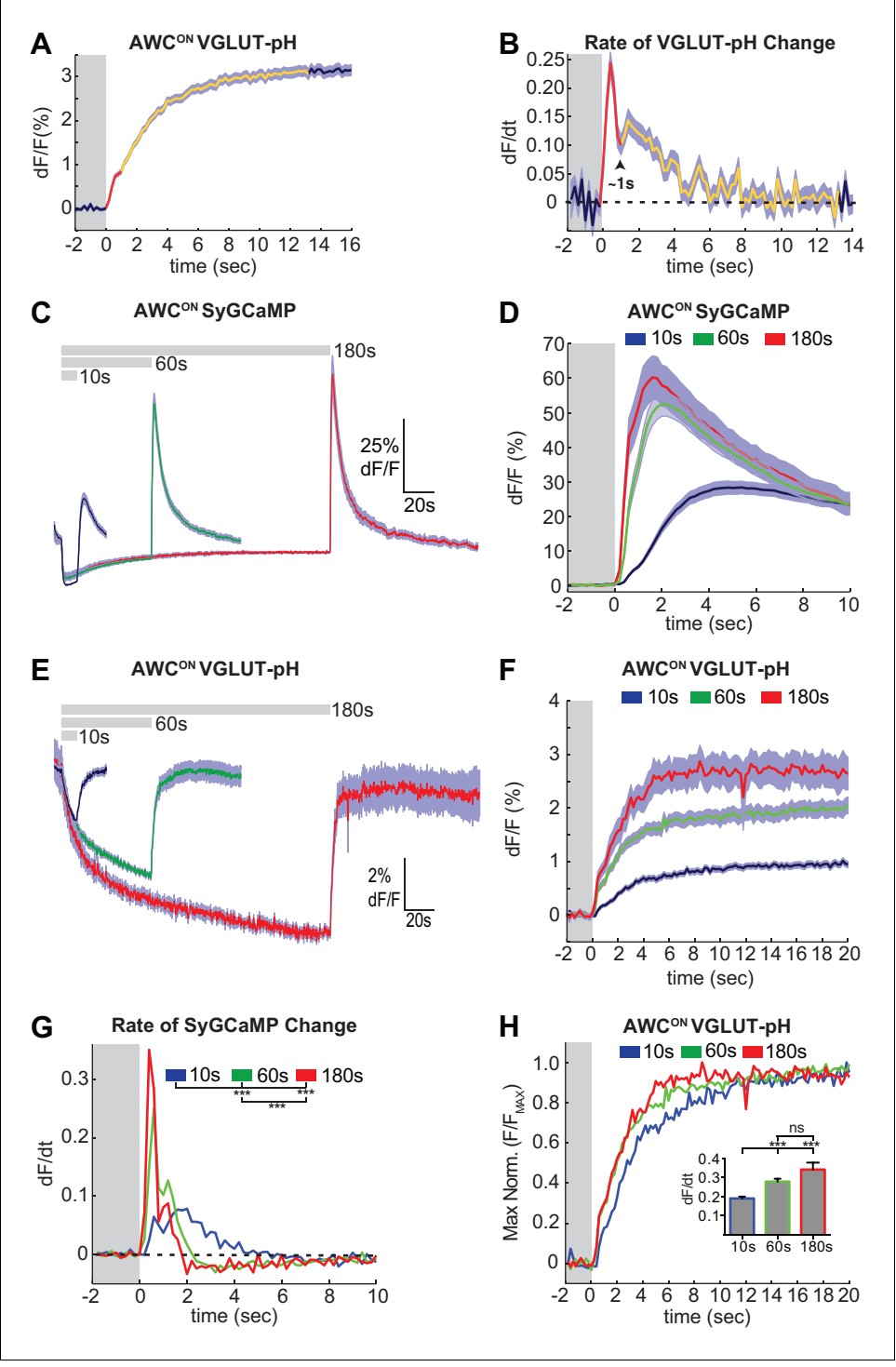

**Figure 2.** Two kinetic phases of VGLUT-pH responses and calcium influx in AWC[ON]. (A) Mean AWC[ON] Vglut-pH signal upon odor removal (60 s stimulus, n = 219 trials, 1–3 trials per animal). Trace is colored according to transitions in time derivative in (B). (B) Mean time derivative of AWC[ON] VGLUT-pH signals in (A). (C) AWC[ON] synaptic calcium responses to butanone pulses measured with syGCaMP; traces aligned to odor addition. 10 s pulses: n = 42 (7 animals, six trials each). 60 s pulses: n = 21 (7 animals, three trials each). 3 min pulses: n = 7 (7 animals, one trial each). (D) syGCaMP responses from (C) aligned to odor removal. (E) AWC[ON] VGLUT-pH responses to butanone pulses; traces aligned to odor addition. 10 s pulses: n = 120 (20 animals, six trials each). 60 s pulses: n = 59 (21 animals, 2–3 trials each). 3 min pulses: n = 14 (14 animals, one trial each). (F) VGLUT-pH responses from (E) aligned to odor removal. (G) Mean time derivative of AWC[ON] syGCaMP signals in (D) shows

*Figure 2 continued on next page*

*Figure 2 continued*

different peak rates 1 s after odor removal (***p<0.0001, one-way ANOVA with Tukey's correction). (**H**) Average VGLUT-pH odor removal responses from (**F**) after normalizing response magnitude. Inset: Average peak time derivative of AWC<sup>ON</sup> VGLUT-pH signals 1 s after odor removal. ***p<0.0001, ns (p=0.14), One-way ANOVA with Tukey's correction. For time derivative plots each individual trial was smoothed with a running average (three frames) before taking the derivative. Units are change in dF/F (%) per 200 ms. Gray areas in A,B,D,F-H mark odor stimulus periods. Shading indicates S.E.M.

DOI: https://doi.org/10.7554/eLife.31234.006

The following source data and figure supplement are available for figure 2:

**Source data 1.** Source data for *Figure 2*.

DOI: https://doi.org/10.7554/eLife.31234.008

**Figure supplement 1.** Time derivative of AWC<sup>ON</sup>and ASH GCaMP signals in the cell body and ASH GCaMP signals in the axon.

DOI: https://doi.org/10.7554/eLife.31234.007

## The SNARE complex drives tonic and evoked synaptic vesicle release

Synaptic vesicle release is triggered by the SNARE complex, which is composed of the plasma membrane proteins syntaxin and SNAP-25, and the vesicle-associated protein synaptobrevin. Expressing the light chain of tetanus toxin (TeTx), which cleaves synaptobrevin (*Schiavo et al., 1992*), in either AWC$^{ON}$ or ASH eliminated their VGLUT-pH responses, as predicted if the VGLUT-pH signals report synaptic vesicle dynamics (*Figure 3A,B*). Notably, AWC$^{ON}$ VGLUT-pH fluorescence did not decrease upon odor addition in TeTx animals, suggesting that both tonic AWC$^{ON}$ exocytosis and the evoked exocytosis after long odor stimuli require the SNARE complex.

Mutant analysis supported the roles of SNARE-complex proteins in sensory exocytosis. Null mutations in *C. elegans* SNARE-complex proteins are inviable, but partial loss of function in syntaxin (*unc-64*) and SNAP-25 (*ric-4*) are viable, with reduced synaptic vesicle release at the neuromuscular junction (*Liu et al., 2005*; *Martin et al., 2011*). Both *unc-64(e246)* (*Figure 3C,D*) and *ric-4(md1088)* (*Figure 3E,F*) had diminished VGLUT-pH responses in AWC$^{ON}$ and ASH. These results are consistent with a requirement for the SNARE complex in tonic and evoked glutamate release from AWC$^{ON}$ and evoked glutamate release from ASH.

## SNARE regulators can differentially affect AWC$^{ON}$ and ASH

A suite of conserved presynaptic proteins including SNARE-associated proteins, scaffold proteins, and small GTPases affect synaptic release in many animals, but their apparent importance varies between reports. Among these presynaptic regulators are UNC-13 and UNC-18 (*Augustin et al., 1999*; *Varoqueaux et al., 2002*; *Verhage et al., 2000*). UNC-13 is implicated in priming synaptic vesicles prior to release; it is a multidomain protein with a MUN domain that can open a closed conformation of syntaxin, and three C2 domains, which bind phorbol esters, phospholipids, and in some cases calcium (*Richmond et al., 1999*). ASH neurons had no detectable VGLUT-pH response to sensory stimuli in *unc-13* null mutants, indicating an absolute requirement for this protein in ASH synaptic vesicle mobilization (*Figure 4B*). By contrast, AWC$^{ON}$ neurons in *unc-13* null mutants had a significant, albeit reduced, increase in VGLUT-pH signal after odor removal indicative of residual synaptic activity (*Figure 4A*).

UNC-18 also interacts with syntaxin, and regulates syntaxin localization as well as activity (*McEwen and Kaplan, 2008*; *Ogawa et al., 1998*). AWC$^{ON}$ VGLUT-pH responses were nearly eliminated in *unc-18* null mutants (*Figure 4C*). However, *unc-18* ASH neurons responded to stimuli by mobilizing VGLUT-pH, albeit to a lesser extent than the wild-type (*Figure 4D*). These results reveal heterogeneity in the requirements for SNARE regulators in different cell types: ASH has a stronger requirement for *unc-13* and a weaker requirement for *unc-18* than AWC$^{ON}$ (Statistics in Supplementary file 3B).

Synaptic vesicle release was reduced but not eliminated in both ASH and AWC$^{ON}$ neurons by mutations affecting the scaffold protein UNC-10/RIM (*Figure 4E,F*). ASH neurons showed enhanced synaptic vesicle release in mutations for the SNARE regulator *cpx-1/complexin* mutants (*Figure 4H*); a similar trend in AWC$^{ON}$ was not statistically significant (*Figure 4G*, *Supplementary file 3B*).

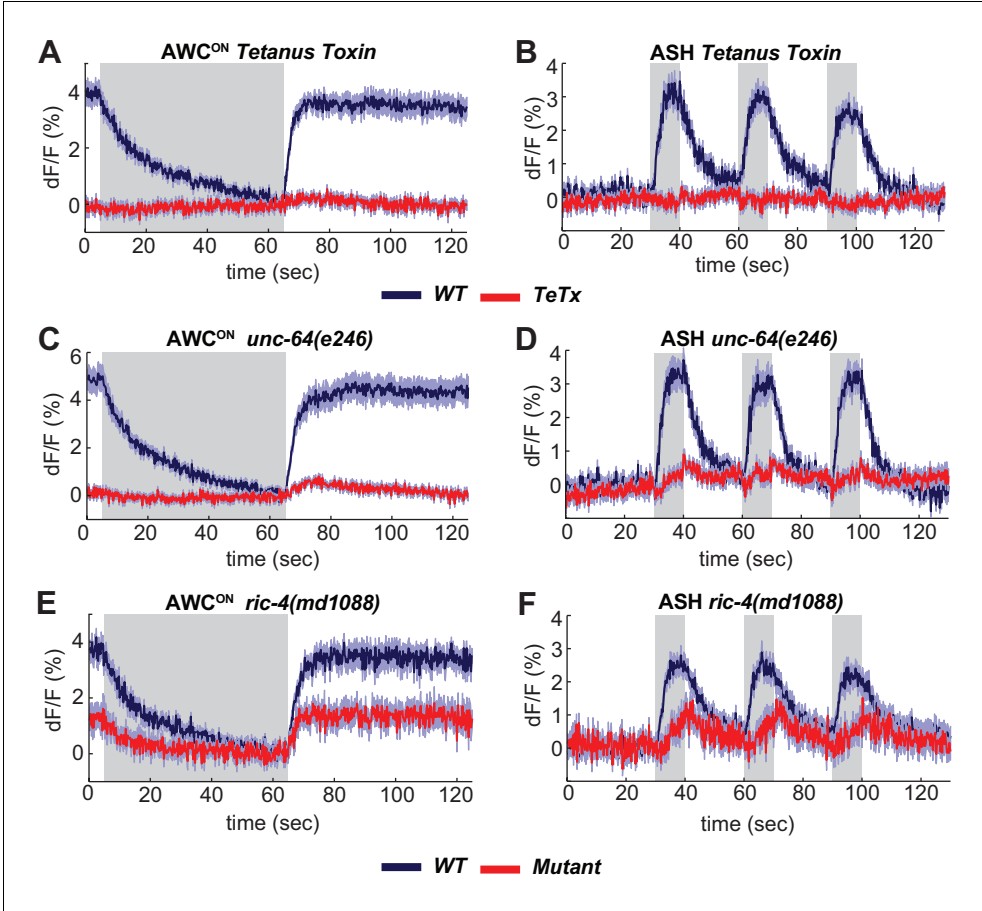

**Figure 3.** The SNARE complex is required for VGLUT-pH responses in AWC[ON] and ASH. (**A,B**) VGLUT-pH responses are eliminated by cell-specific expression of TeTx light chain. (**A**) AWC[ON] *str-2* promoter driving TeTx n = 25 (9 animals, 2–3 trials each). AWC[ON] *wt* (TeTx-array negative animals tested in parallel) n = 28 (10 animals, 2–3 trials each). (**B**) ASH *sra-6* promoter driving TeTx n = 33 (11 animals, three trials each). ASH *wt* (TeTx-array negative animals tested in parallel) n = 24 (8 animals, three trials each). (**C,D**) VGLUT-pH responses in *unc-64(e246)* (partial loss of function) syntaxin mutants. (**C**) AWC[ON] *unc-64(e246)* n = 30 (10 animals, three trials each). AWC[ON] *wt* n = 23 (9 animals, 2–3 trials each). (**D**) ASH *unc-64(e246)* n = 24 (8 animals, three trials each). ASH *wt* n = 18 (6 animals, three trials each). (**E, F**) VGLUT-pH responses in *ric-4(md1088)* (partial loss of function) SNAP-25 mutants. (**E**) AWC[ON] *ric-4(md1088)* n = 12 (4 animals, three trials each). AWC[ON] *wt* n = 15 (5 animals, three trials each). (**F**) ASH *ric-4(md1088)* n = 16 (6 animals, 2–3 trials each). ASH *wt* n = 32 (11 animals, 2–3 trials each). Mutations are described in *Supplementary file 2*. All differences are significant (p<0.0001, unpaired t-test), as detailed in *Supplementary file 3A*. Gray areas mark stimulus periods. Shading indicates S.E.M.
DOI: https://doi.org/10.7554/eLife.31234.009

The following source data is available for figure 3:

**Source data 1.** Source data for *Figure 3*.
DOI: https://doi.org/10.7554/eLife.31234.010

## Cytoplasmic pH is increased by neuronal activity, independent of synaptic release

As a counterpoint to the VGLUT-pHluorin experiments, we examined the effect of sensory stimuli on cytoplasmic pH in AWC[ON] and ASH neurons. Activity-dependent increases or decreases in cytoplasmic pH have been documented in both vertebrate and invertebrate neurons (*Chesler, 2003*; *Rossano et al., 2013*; *Zhang et al., 2010*). Similarly, expressing an untagged super-ecliptic pHluorin in the cytoplasm of AWC[ON] and ASH neurons (cyto-pH) reported robust stimulus-dependent pH changes. Odor addition increased cyto-pH fluorescence in AWC[ON], suggesting alkalinization, and odor removal decreased cyto-pH fluorescence, suggesting activity-dependent acidification

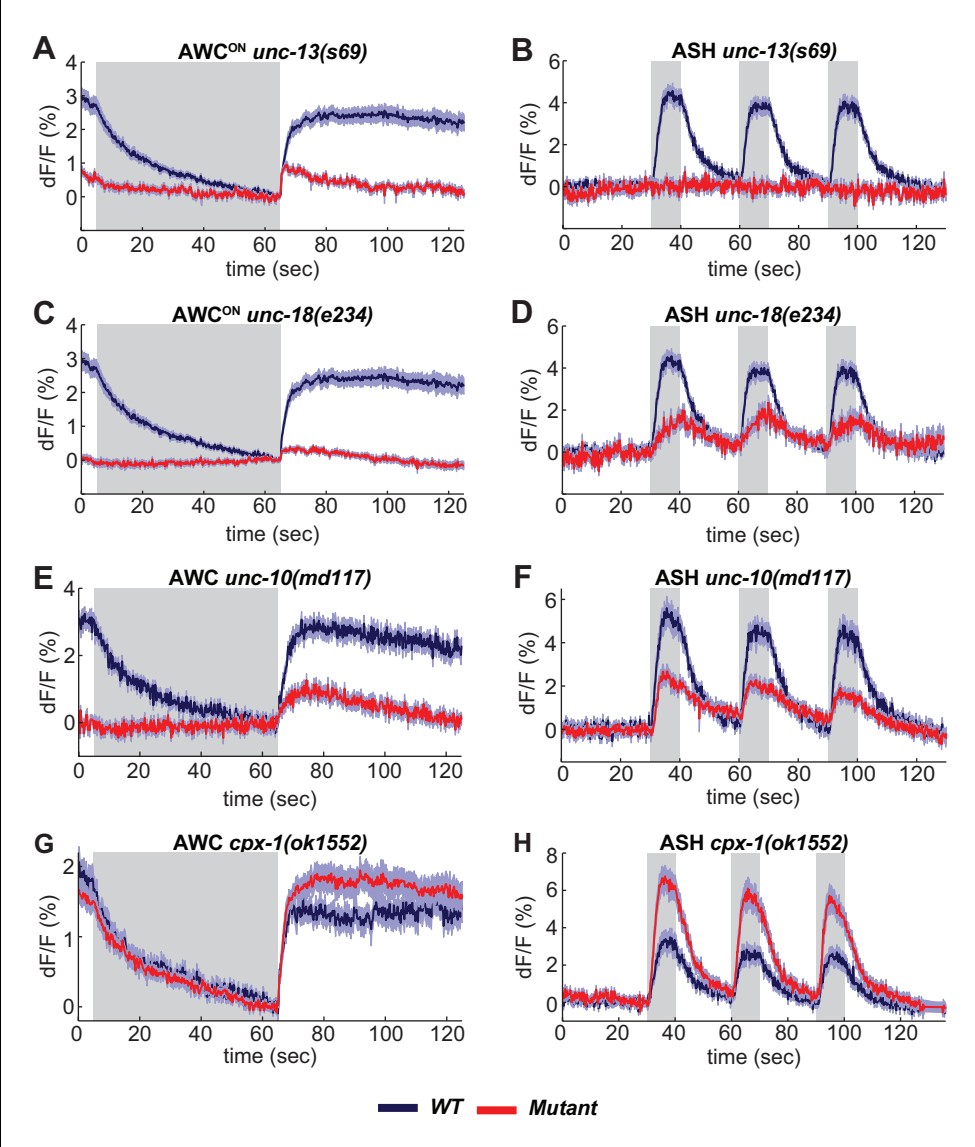

**Figure 4.** SNARE regulators differentially affect AWC[ON] and ASH. (**A,B**) VGLUT-pH responses in *unc-13(s69)* null mutants. (**A**) AWC[ON] responses in *unc-13(s69)*, n = 25 (9 animals, 2–3 trials each) and *wt*, n = 33 (11 animals, three trials each). (**B**) ASH responses in *unc-13(s69)*, n = 18 (7 animals, 1–3 trials each) and *wt*, n = 42 (14 animals, three trials each). (**C,D**) VGLUT-pH responses in *unc-18(e234)* mutants. (**C**) AWC[ON] responses in *unc-18(e234)*, n = 25 (9 animals, 1–3 trials each) and *wt*, n = 33 (11 animals, three trials each). (**D**) ASH responses in *unc-18(e234)*, n = 22 (8 animals, 1–3 trial each) and *wt*, n = 42 (14 animals, three trials each). (**E,F**) VGLUT-pH responses in *unc-10(md117)* mutants. (**E**) AWC[ON] responses in *unc-10(md117)*, n = 23 (8 animals, 1–3 trials each), and *wt*, n = 27 (9 animals, three trials each). (**F**) ASH responses in *unc-10(md117)*, n = 33 (12 animals, 2–3 trials each), and *wt*, n = 24 (8 animals, three trials each). (**G,H**) VGLUT-pH responses in *cpx-1(ok1552)* mutants. (**G**) AWC[ON] responses in *cpx-1(ok1552)*, n = 36 (12 animals, three trials each), and *wt*, n = 17 (6 animals, 2–3 trials each). (**H**) ASH responses in *cpx-1(ok1552)*, n = 38 (13 animals, 2–3 trials each), and *wt* n = 33 (11 animals, three trials each). WT and mutants are significantly different in panels A-F and H, as detailed in **Supplementary file 3B**. Gray areas mark stimulus periods. Shading indicates S.E.M.
DOI: https://doi.org/10.7554/eLife.31234.011

The following source data is available for figure 4:

**Source data 1.** Source data for **Figure 4**.
DOI: https://doi.org/10.7554/eLife.31234.012

(*Figure 5A*). These pH changes were opposite in sign to the signals defected by VGLUT-pH (compare *Figure 5A* to 1G), and were strongest in the axon, intermediate in the cell body, and weak in the sensory dendrite (*Figure 5A*). In ASH, NaCl stimuli elicited a decrease in cyto-pH fluorescence suggestive of acidification, again the opposite sign of the VGLUT-pH signal (*Figure 5B*).

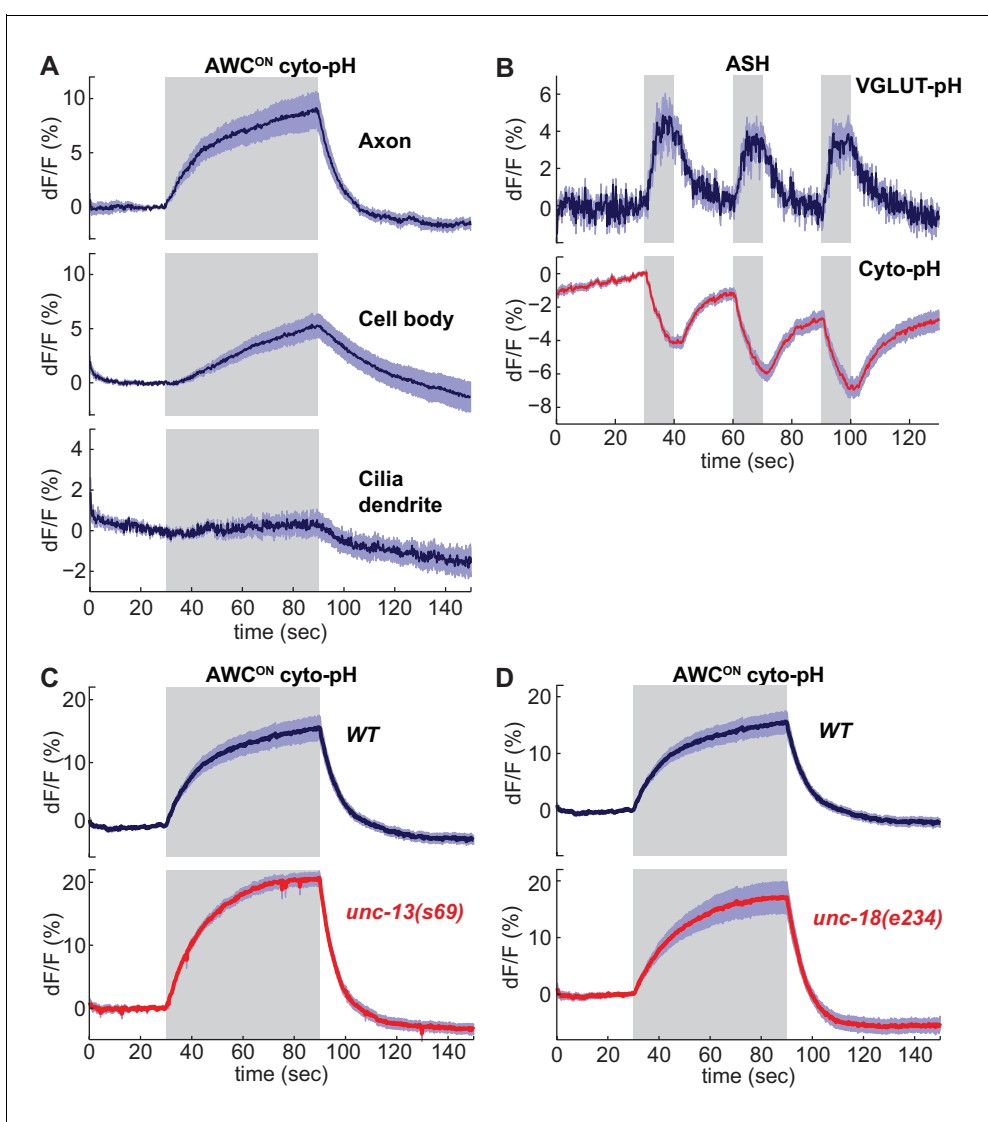

**Figure 5.** Sensory stimuli evoke cytoplasmic pH changes in AWC[ON] and ASH. (A) Average cyto-pH signals at different subcellular sites of AWC[ON]. Axon and cell body, n = 9 (3 animals, three trials each), Cilia-dendrite n = 6 (2 animals, three trials each). (B) Average ASH VGLUT-pH response (top) and cyto-pH response (bottom), tested in parallel under the same stimulus conditions. VGLUT-pH n = 9 (3 animals, three trials each). cyto-pH n = 27 (9 animals, three trials each). (C,D) Average AWC[ON] cyto-pH responses in (C) *unc-13(s69)* and (D) *unc-18(e234)* mutants. Mutants and wild-type controls were measured on the same days; neither mutant was significantly different from wild-type (p>0.12, one-way ANOVA with Tukey's correction). *unc-13(s69)* n = 15 (5 animals, three trials each). *unc-18(e234)* n = 11 (4 animals, 2–3 trials each). *wt* n = 18 (6 animals, three trials each). Gray areas mark stimulus periods. Shading indicates S.E.M.

DOI: https://doi.org/10.7554/eLife.31234.013
The following source data is available for figure 5:

**Source data 1.** Source data for *Figure 5*.
DOI: https://doi.org/10.7554/eLife.31234.014

Unlike VGLUT-pH responses, AWC^ON cyto-pH signals were normal in *unc-13* or *unc-18* mutations (*Figure 5C,D*). These results suggest that cytoplasmic pH changes are independent of synaptic vesicle release.

## The endocytosis-reacidification process is accelerated by AP180/pCALM

The decrease in VGLUT-pH fluorescence at synapses represents the recapture of the protein from the cell surface by endocytosis and the acidification of the resulting synaptic vesicles (*Balaji and Ryan, 2007*; *Sankaranarayanan and Ryan, 2000*)(*Figure 1—figure supplement 1C*). To estimate the rate of this combined retrieval step, we fit fluorescence decreases from individual AWC^ON and ASH trials to an exponential decay function (*Figure 6A,B*) (*Smith et al., 2008*). A large fraction of traces were consistent with single exponential decay, with an 18 s time constant for AWC^ON and an 8 s time constant for ASH (*Figure 6C,D*); a subset of traces were consistent with double exponential decay (*Figure 6—figure supplement 1*). The measured decay constants within and between neurons did not correlate with axon fluorescence before endocytosis (*Figure 6—figure supplement 2*), suggesting that VGLUT-pH expression levels did not saturate the endocytosis machinery (*Sankaranarayanan and Ryan, 2000*).

Among the proteins most strongly implicated in synaptic vesicle retrieval is the adaptor protein AP180, which clusters synaptic vesicle proteins (*Gimber et al., 2015*; *Koo et al., 2011*) and interacts with AP-2/clathrin at a sorting stage immediately after endocytosis (*Koo et al., 2015*). The *C. elegans* AP180/CALM homolog *unc-11* has long been proposed to affect endocytosis, as well as affecting synaptic vesicle morphology and protein sorting (*Nonet et al., 1999*). Indeed, in AWC^ON neurons upon odor addition, and in ASH neurons after stimulus removal, *unc-11(e47)* null mutants had significantly slowed VGLUT-pH retrieval (*Figure 6E–H*). Exocytosis may also be impacted in *unc-11(e47)* mutants, as a significant fraction of trials produced weak or undetectable responses in ASH (*Figure 6F*, gray trace). Baseline VGLUT-pH fluorescence was higher in *unc-11(e47)* mutants than in wild-type for both neurons, consistent with increased VGLUT-pH on the cell surface or in other neutral compartments (*Figure 6I,J*).

## Activity-dependent regulation of VGLUT-pH retrieval in AWC^ON

The distinct endocytosis and recapture rates in AWC^ON and ASH could reflect either cell type-specific or cell state-specific processes. With respect to cell state, the GCaMP signals in AWC^ON and ASH suggest that the calcium levels in ASH at the end of a NaCl stimulus resemble those in AWC^ON after odor removal, not the basal calcium levels when odor is added (*Figure 1A,B*; see Materials and methods). This difference in calcium levels could affect vesicle traffic, as endocytosis in other systems is accelerated at high calcium concentrations (*Leitz and Kavalali, 2016*; *Neves et al., 2001*; *Sankaranarayanan and Ryan, 2001*). To separate the effects of cell type and cell state, we compared VGLUT-pH retrieval in AWC^ON at basal and elevated calcium levels. Odor was delivered to AWC^ON, removed after one minute to elicit a calcium overshoot, and then delivered again after 10 s while calcium levels were still elevated (*Figure 7A*). Strikingly, AWC^ON VGLUT-pH retrieval was accelerated during the second odor pulse, matching the ~8 s retrieval time observed in ASH (*Figure 7B,C*). The effect was temporary (lasting <70 s) but could be induced again after another 60 s odor pulse (*Figure 7D*). *unc-11(e47)* mutants were also regulated by the dual odor-pulse protocol, and delayed compared to wild-type under both conditions (*Figure 7E,F*). These results indicate that VGLUT-pH retrieval in AWC^ON is regulated by activity, consistent with calcium-dependent acceleration of endocytosis.

## Protein kinase C epsilon regulates AWC^ON exocytosis downstream of calcium

As a first step toward using VGLUT-pH imaging to examine more selective regulators of synaptic transmission, we examined mutants in the protein kinase C epsilon (novel class) encoded by *pkc-1*. *pkc-1* is required for normal behavioral responses to odors detected by AWC^ON, and has previously been suggested to act at a step downstream of AWC^ON calcium entry (*Tsunozaki et al., 2008*). We confirmed the normal calcium response to odor stimulation in *pkc-1* mutants (*Figure 8—figure supplement 1A,B*), and additionally determined that synaptic calcium signals detected with

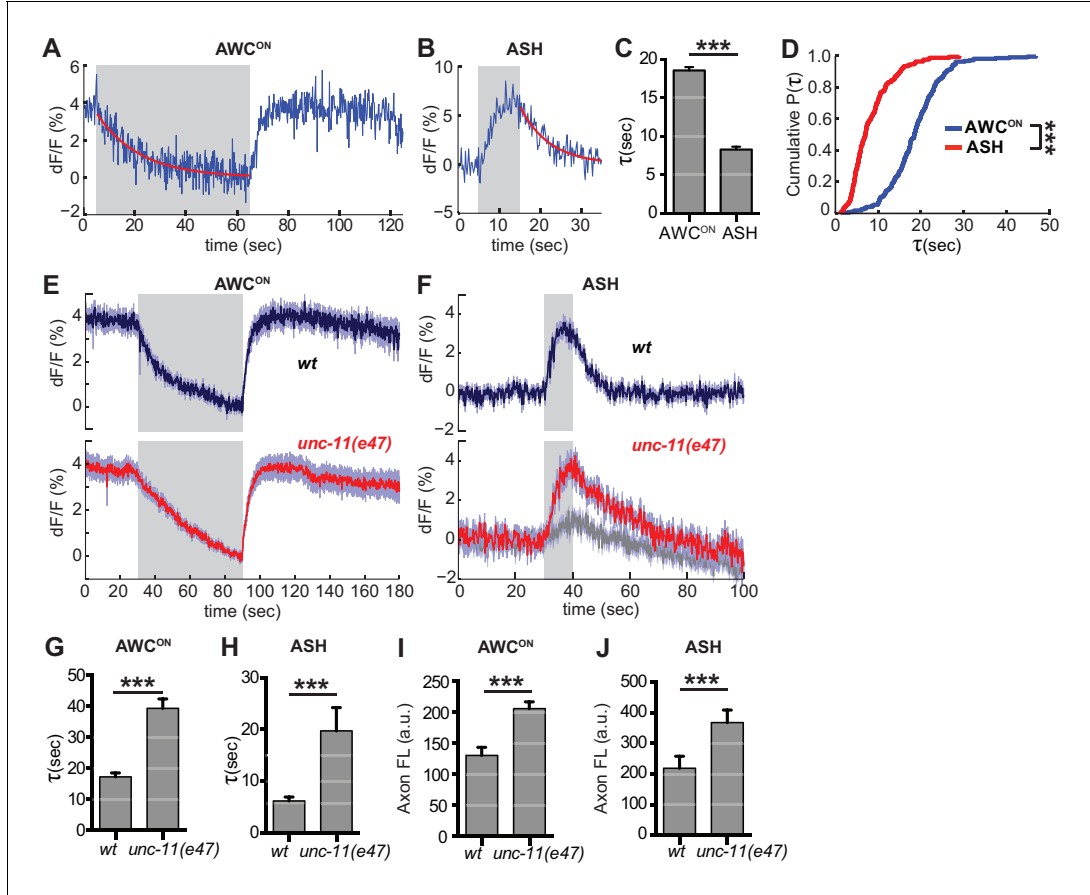

**Figure 6.** Synaptic vesicle retrieval is accelerated by AP180/CALM. (**A,B**) Representative single exponential fits (red) to single trials of (**A**) AWC[ON] and (**B**) ASH VGLUT-pH decays upon stimulus addition or removal, respectively. For each neuron, some responses were consistent with double exponential decay models (*Figure 6—figure supplement 1*). (**C**) Average time constant of AWC[ON] and ASH decays from single exponential fits. AWC[ON] n = 218 (76 animals, 2–3 trials each). ASH n = 168 (56 animals, 2–3 trials each). ***p<0.0001, unpaired t-test. (**D**) Empirical cumulative distribution plot of data in (**C**). Distributions differ by Kolmogorov-Smirnov test, ***p<0.0001. (**E,F**) Average VGLUT-pH responses in *unc-11(e47)* mutants. (**E**) AWC[ON] responses in *unc-11(e47)* n = 27 (10 animals, 2–3 trials each), *wt* n = 15 (5 animals, three trials each). One *unc-11(e47)* animal did not respond and was removed from the analysis. (**F**) ASH responses in *unc-11(e47)* mutants. *wt* n = 21 (7 animals, three trials each). *unc-11(e47)* n = 23 (8 animals, 2–3 trials each). Red trace: mean of 9 trials (five animals, 1–2 trails each) with clear responses to odor addition. Magnitude of response does not differ from *wt* (p=0.72, peak odor response); Gray trace: mean of 14 trials that produced weak or non-detectable responses to odor addition, significantly different from *wt* (p<0.0001, peak odor response). One-way ANOVA, Tukey's correction. (**G,H**) Average time constants from single exponential fits (initial 20 s of decay) of data in (**E, F**). For ASH *unc-11(e47)* mutants, only data from the red trace was used. Unpaired t-test, p<0.0001. AWC[ON] *unc-11(e47)* n = 25 (10 animals, 2–3 trials each); *wt* n = 15 (5 animals, three trials each). ASH n as in (**F**). (**I,J**) Average axon fluorescence (first 5 frames of the recording). (**I**) AWC[ON] *wt* n = 12 animals, *unc-11(e47)* n = 19 animals. (**J**) ASH *wt* n = 7 animals, *unc-11(e47)* n = 8 animals. Unpaired t-test, p<0.0001. Gray areas mark stimulus periods. Shading and error bars indicate S.E.M.

DOI: https://doi.org/10.7554/eLife.31234.015

The following source data and figure supplements are available for figure 6:

**Source data 1.** Source data for *Figure 6*.
DOI: https://doi.org/10.7554/eLife.31234.018

**Figure supplement 1.** A proportion of VGLUT-pH recordings are consistent with multiple retrieval time constants.
DOI: https://doi.org/10.7554/eLife.31234.016

**Figure supplement 2.** VGLUT-pH decay time constants are not correlated with expression level or response magnitude.
DOI: https://doi.org/10.7554/eLife.31234.017

synaptogyrin-GCaMP were normal or slightly increased in *pkc-1* mutants (*Figure 8A,B*, top). By contrast, VGLUT-pH signals in mutants with the kinase-inactivating mutation *pkc-1(nj1)* were severely diminished across all conditions, resembling those of strong SNARE mutants (*Figure 8B,C*). Three additional *pkc-1* alleles had related defects: in each case, the response to odor addition was nearly

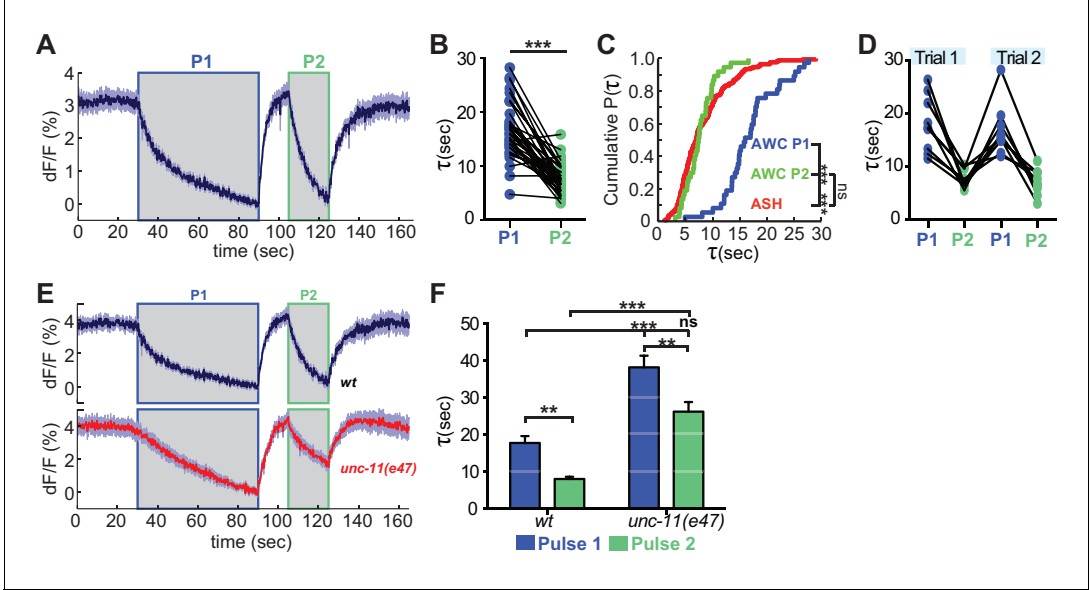

**Figure 7.** Recent neural activity modulates AWC[ON] VGLUT-pH retrieval. (**A**) Average AWC[ON] VGLUT-pH response to two successive odor stimuli, applied for 60 s (**P1**) and 20 s (**P2**). n = 39 (13 animals, three trials each). (**B**) Time constants from single-term exponential fits to P1 and P2 from (**A**) performed on the initial 20 s of the decay for each pulse. n = 37 (13 animals, 2–3 trials each). Paired t-test, ***p<0.0001. (**C**) Cumulative distribution plot of time constants for AWC P1, AWC P2, and ASH VGLUT-pH decays. AWC P1 and AWC P2 data from (**B**) and ASH data from *Figure 6D*. Kruskal-Wallis and Dunn's test for multiple comparisons, ***p<0.0001, ns p=0.1. (**D**) Time constants for P1 and P2 from two consecutive trials of the stimulation protocol in (**A**) (n = 8 animals, 70 s between trials). (**E**) Average AWC VGLUT-pH signals in *wt* and *unc-11(e47)* mutants. *wt* n = 21 (7 animals, three trials each). *unc-11(e47)* n = 25 (9 animals, three trials each) (two non-responding trials removed). (**F**) Average time constants from single exponential fits (initial 20 s of decay) of data in (**E**). *wt* n = 21 (7 animals, three trials each). *unc-11(e47)* n = 22 (8 animals, 2–3 trials each). Two-way ANOVA, ***p<0.0001, **p<0.008, ns (p=0.07). Gray areas marks stimulus periods. Shading and error bars indicate S.E.M. ns = not significant.

DOI: https://doi.org/10.7554/eLife.31234.019

The following source data is available for figure 7:

**Source data 1.** Source data for *Figure 7*.
DOI: https://doi.org/10.7554/eLife.31234.020

eliminated, and the response to odor removal was diminished to a greater or lesser degree (*Figure 8—figure supplement 1D*).

Selective transgenic expression of a *pkc-1* cDNA in AWC[ON] resulted in full rescue of the VGLUT-pH defect and the behavioral defect (*Figure 8D*, *Figure 8—figure supplement 1C*), indicating that *pkc-1* affects synaptic vesicle exocytosis cell autonomously in AWC[ON].

In contrast with AWC[ON], VGLUT-pH signals in ASH neurons were only slightly affected by *pkc-1*: *pkc-1* mutants substantially preserved both exocytosis and retrieval dynamics in ASH (*Figure 8E–G*). A subtle *pkc-1* defect was observed upon repetitive stimulation, where VGLUT-pH exocytosis responses were diminished compared to wild-type (*Figure 8F*). This effect was temporary and recovered within ~1 min.

## Discussion

Physiological sensory stimuli elicit different patterns of synaptic vesicle exocytosis and retrieval in the AWC[ON] and ASH sensory neurons, as inferred from real-time changes in VGLUT-pH fluorescence. The ASH neuron has two distinct synaptic states: a basal state with low-calcium levels and low exocytosis, and a stimulated state with high calcium levels and increased exocytosis. The AWC[ON] neuron has three states: a basal state with tonic exocytosis and retrieval, an odor-evoked low-calcium state in which exocytosis is suppressed but retrieval continues, and a transient state after the removal of a long-duration odor stimulus with high calcium, accelerated exocytosis, and accelerated endocytosis.

VGLUT-pH signals in AWC[ON] and ASH are correlated with sensory-evoked calcium dynamics across a variety of conditions, and their sensitivity to genetic manipulations of the conserved SNARE

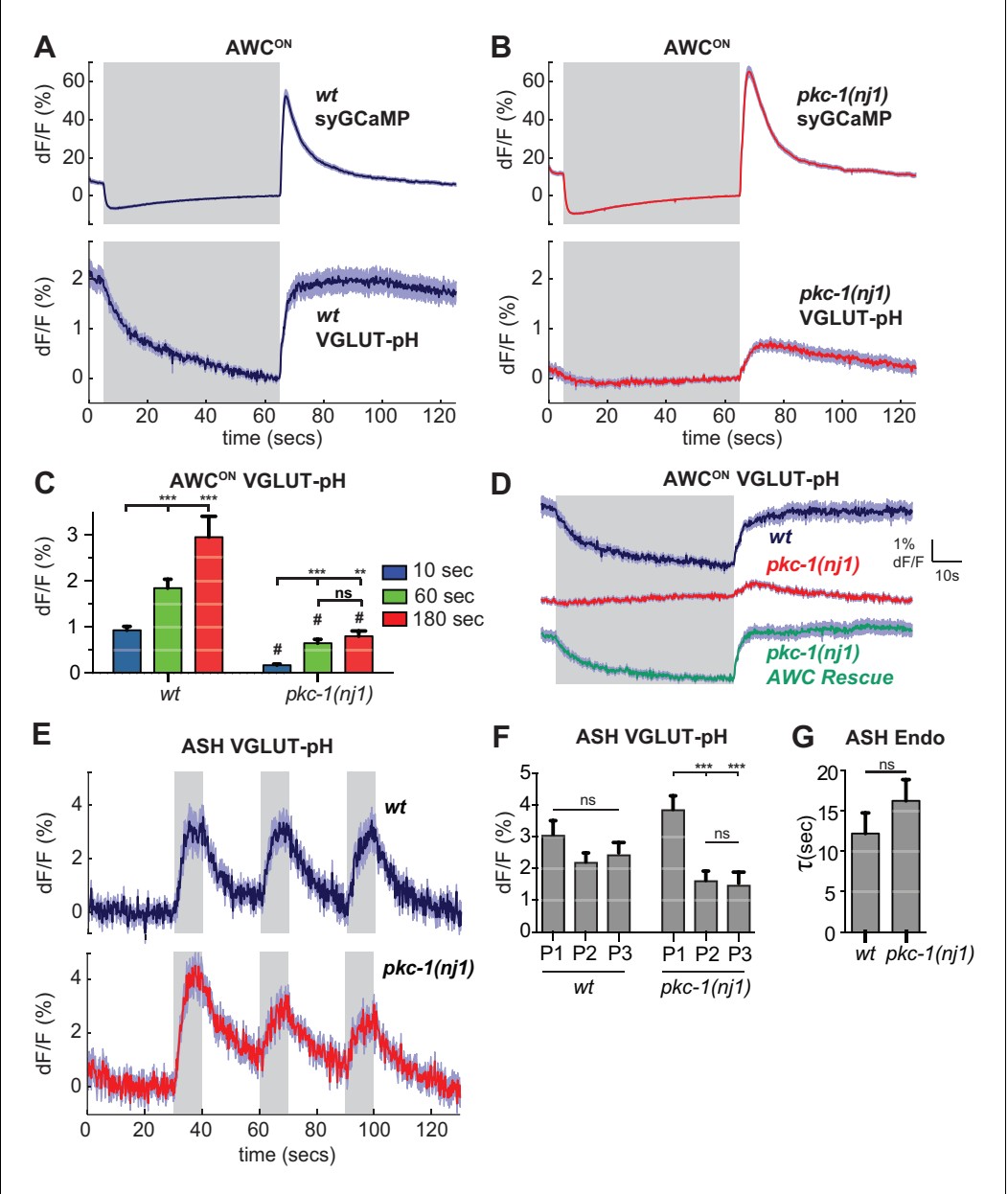

**Figure 8.** *pkc-1* regulates AWC[ON] glutamate release downstream of calcium influx. (A,B) AWC[ON] synaptic calcium (top) and VGLUT-pH (bottom) responses in (A) wild-type and (B) *pkc-1(nj1)* mutant animals. *wt* SyGCaMP n = 21 (7 animals, three trials each), *pkc-1* SyGCaMP n = 25 (9 animals, 1–3 trials each). *wt* VGLUT-pH n = 29 (10 animals, 2–3 trials each), *pkc-1(nj1)* VGLUT-pH n = 48 (17 animals, 2–3 trials each). (C) Average AWC[ON] VGLUT-pH peak response magnitude after odor removal for indicated stimulation durations. #, different from wild-type p<0.0001. **p=0.0026, ***p<0.0001. Two-way ANOVA Tukey's correction. 10 s pulses: *wt* n = 60 (10 animals, six trials each). *pkc-1(nj1)* n = 102 (17 animals, 4–6 trials each). 60 s pulses: as in (A,B). 3 min pulses: *wt* n = 8 (8 animals, one trial each). *pkc-1(nj1)* n = 15 (15 animals, one trial each). (D) Expression of *pkc-1* cDNA in AWC[ON] rescues *pkc-1(nj1)* VGLUT-pH responses. *wt* n = 18 (6 animals, three trials each). *pkc-1(nj1)* n = 30 (10 animals, three trials each). *pkc-1 (nj1)* AWC[ON] rescue n = 30 (10 animals, three trials each). (E) ASH VGLUT-pH responses in *pkc-1(nj1)* mutants. *wt* n = 15 (5 animals, three trials each). *pkc-1(nj1)* n = 15 (6 animals, 2–3 trials each). (F) Average ASH VGLUT-pH peak responses for each stimulus pulse (P1-3) within a trial. Data from (E). Two-way ANOVA Tukey's correction. ***p<0.0001. (G) Average time constants from single exponential fits of data in (E, first pulse). Supporting statistical analysis for all panels is detailed in *Supplementary file 3C* and Source Data. Gray areas mark stimulus periods. Shading and error bars indicate S.E.M. ns = not significant.

DOI: https://doi.org/10.7554/eLife.31234.021

*Figure 8 continued on next page*

*Figure 8 continued*

The following source data and figure supplement are available for figure 8:

**Source data 1.** Source data for *Figure 8*.

DOI: https://doi.org/10.7554/eLife.31234.023

**Figure supplement 1.** *pkc-1(lf)* alters synaptic release downstream of calcium influx.

DOI: https://doi.org/10.7554/eLife.31234.022

proteins and regulators of endocytosis support the conclusion that they reflect endogenous synaptic vesicle dynamics. The graded properties inferred for AWC[ON] neurons are consistent with those of *C. elegans* motor neurons measured at the neuromuscular junction (*Liu et al., 2009*), in agreement with the absence of sodium-based action potentials in nematode neurons (*Goodman et al., 1998*; *Liu et al., 2009*). Another point of similarity between AWC[ON] and motor neurons is the existence of distinct basal and evoked synaptic release modes (*Richmond et al., 1999*; *Francis et al., 2005*; *Martin et al., 2011*), which may differentially transmit signals to downstream behavioral circuits (*Kato et al., 2014*).

## Cell type-specific regulation of the synaptic vesicle machinery in sensory neurons

The SNARE complex is required at all known synapses, but the detailed functions of the highly conserved SNARE regulators are still being determined. We found that the requirements for SNARE regulators differed between neurons. Both AWC[ON] and ASH had a partial requirement for *unc-10/ RIM. unc-13* was essential for all VGLUT-pH responses in ASH, in agreement with previous studies of postsynaptic responses to ASH stimulation (*Lindsay et al., 2011*), whereas *unc-18* was less important. By contrast, AWC[ON] was more strongly dependent on *unc-18* than *unc-13*. Thus the synaptic requirement for *unc-13* and *unc-18* may differ across different synapses or conditions, even in cells in which both genes are expressed and active (*Atwood and Karunanithi, 2002*; *Crawford and Kavalali, 2015*; *Kasai et al., 2012*)

In *unc-13* mutants, AWC[ON] did not respond to odor addition with a normal decrease in VGLUT-pH, but did exhibit some exocytosis immediately after odor removal. These dynamics suggest that tonic AWC[ON] release cannot be maintained in *unc-13(lf)* mutants, but synaptic vesicles can be released after a large calcium influx. This result, among others, suggests that the underlying difference between ASH and AWC[ON] cannot be explained entirely by the tonic-evoked distinction: in ASH, *unc-13* is required for evoked release, but in AWC[ON], it is more important in tonic exocytosis.

Complexin interacts with the SNARE proteins synaptobrevin and syntaxin, modifying their function to inhibit basal release and facilitate evoked release (*Malsam et al., 2008*; *Martin et al., 2011*; *McMahon et al., 1995*; *Rizo and Xu, 2015*; *Wragg et al., 2013*; *Xue et al., 2009*). The striking increase in ASH VGLUT-pH signals in *cpx-1(lf)* mutants, which was not expected based on previous complexin studies, points to another difference between ASH and AWC[ON]. The sources of these differences might reside in cell type-specific calcium channels, synaptic vesicle pools, or endocytosis machinery.

Many questions about synaptic vesicle dynamics in wild-type and mutant neurons remain to be addressed. We have not examined the roles of different calcium sources in AWC[ON] and ASH, although previous studies have demonstrated that both CaV2 *unc-2* channels and CaV1 *egl-19* channels contribute to calcium entry at the neuromuscular junction (*Tong et al., 2017*), and both channels are also implicated in the signaling functions of AWC[ON] and ASH (*Saheki and Bargmann, 2009*; *Zahratka et al., 2015*). The full relationship between the VGLUT-pH vesicle pools and releasable synaptic vesicle pools is also unknown, although it is encouraging that the VGLUT-pH signals are similar to those observed in biophysically well-characterized systems (*Ariel and Ryan, 2010*). Finally, for all mutants except *pkc-1*, we have not determined whether the defects are intrinsic to AWC[ON] and ASH, and therefore it is possible that some of the phenotypes described here are indirect consequences of gene action in other cells, large-scale circuit changes, or altered neuromodulation.

## The PKC-1 protein kinase C homolog regulates synaptic vesicle release in AWC[ON]

A striking difference between AWC[ON] and ASH was their dependence on the protein kinase C epsilon homolog PKC-1. *pkc-1* mutants had severe defects in VGLUT-pH mobilization in AWC[ON], with a near complete elimination of the response to odor addition. Different alleles varied in their severity with respect to odor removal, where two alleles that affected the conserved protein kinase domain but spared other protein domains had the strongest phenotypes. These kinase-dead alleles might interfere with interacting proteins, or they might prevent compensation by related kinases whose functions partially overlap with PKC-1 (*Madhani et al., 1997*; *Okochi et al., 2005*).

ASH function was nearly normal in *pkc-1* mutants, although a subtle defect could be uncovered by pulsing ASH with multiple stimuli. Previous studies at the neuromuscular junction indicated that *pkc-1* affects neuropeptide release but not fast synaptic transmission from cholinergic and GABAergic motor neurons (*Sieburth et al., 2007*). Moreover, *pkc-1* mutants have near-normal locomotion, contrasting with the severe locomotion defects in SNARE mutants, providing further evidence of residual fast synaptic transmission at the neuromuscular junction. We suggest that the motor neurons, like ASH, can mobilize synaptic vesicles without *pkc-1,* in contrast with AWC[ON] where *pkc-1* has a substantial role.

Protein kinase C is a well-established potentiator of neurotransmitter release at mammalian synapses (*Wierda et al., 2007*), a result consistent with those observed here. Interestingly, one of the targets of mammalian PKC is the homolog of UNC-18, Munc18-1, paralleling our observation that AWC[ON] exocytosis has a strong requirement for both UNC-18 and PKC-1. In mouse hippocampal neurons, Munc18-1 clustering at synapses is regulated by activity, calcium influx, and protein kinase C phosphorylation, and correlates with synaptic strength (*Cijsouw et al., 2014*). At the mouse Calyx of Held, Munc18-1 PKC phosphorylation sites are important for post-tetanic potentiation, a form of plasticity that enhances neurotransmitter release (*Genc et al., 2014*). *C. elegans* UNC-18 shares consensus PKC phosphorylation sites, which may be phosphorylated by PKC-2, a different PKC, in thermosensory neurons (*Edwards et al., 2012*). PKC has other synaptic targets as well; for example, it phosphorylates the calcium sensor synaptotagmin-1 at hippocampal synapses to potentiate synaptic vesicle release (*de Jong et al., 2016*).

*unc-13* and *unc-18* mutants have severe locomotion defects and cannot be easily tested for sensory behaviors, but the more agile *pkc-1* mutants have been shown to be repelled, rather than attracted, by odors sensed by AWC[ON] (*Tsunozaki et al., 2008*). Their responses to temperature sensed by AFD neurons can also have a reversed valence, with attraction to high temperatures that are normally repulsive (*Luo et al., 2014*; *Okochi et al., 2005*). The behavioral reversal in sensory responses could be related to dynamics of residual glutamate signaling, or it could result from an alternative form of neurotransmission such as neuropeptide release.

## Activity-dependent cytoplasmic pH decreases in glutamatergic neurons

Stimuli that evoke calcium increases in AWC[ON] and ASH result in decreased cytoplasmic pH, as reported by pHluorin fluorescence changes. One potential source of the cytoplasmic change is proton mobilization during calcium extrusion by the plasma membrane calcium ATPase (*Rossano et al., 2013*; *Schwiening and Willoughby, 2002*; *Trapp et al., 1996*; *Zhang et al., 2010*). During periods of elevated calcium levels, pumps are active and acidify the neuron by exchanging protons for calcium ions; when the activity of the neuron decreases and calcium is extruded, activity of the pumps decreases, and the neuron returns to baseline pH levels (*Trapp et al., 1996*). Consistent with calcium-driven acidosis, both calcium dynamics and pH changes were faster in the axon than in the soma.

The stimulus-induced, exocytosis-independent cytoplasmic pH changes we observed in AWC[ON] and ASH are worth future study, and may have unintended technical effects on other imaging experiments. For example, GCaMP fluorescence increases by as much as 10% for a 0.1 pH unit increase at physiological pH (*Kneen et al., 1998*; *Nakai et al., 2001*), and therefore stimulus protocols that alter cytoplasmic pH may confound GCaMP imaging in the same cells.

## Synaptic vesicle retrieval in glutamatergic neurons

In simple stimulus protocols, ASH neurons had a faster apparent rate of synaptic vesicle retrieval – the combination of endocytosis and reacidification – than AWC$^{ON}$ neurons. However, the distribution of AWC$^{ON}$ decay rates was shifted to faster timescales after a previous stimulation with odor (the acceleration effect), producing a distribution similar to that of ASH. The acceleration effect in AWC$^{ON}$ endocytosis may be mediated by the large calcium influx generated by the removal of the first odor pulse, as calcium affects endocytosis in other species (*Leitz and Kavalali, 2016*; *Neves et al., 2001*; *Sankaranarayanan and Ryan, 2001*). Alternative or additional sources of this activity-dependent signal in AWC$^{ON}$ include cGMP, which can accelerate endocytosis (*Bargmann, 2006*; *Petrov et al., 2008*) or cytoplasmic alkalinization (*Zhang et al., 2010*), as the AWC$^{ON}$ axon becomes significantly alkalinized during odor stimulation.

The average basal and accelerated time constants of synaptic vesicle retrieval in AWC$^{ON}$ are ~18 s and ~8 s, within the range reported in other systems. For example, the time constant of endocytosis in hippocampal neurons at physiological temperatures is ~6 s (*Balaji and Ryan, 2007*), and in goldfish retinal OFF-bipolar cells, the time constants for fast and slow endocytosis measured using a capacitance clamp were 1 s and >10 s, respectively (*Neves and Lagnado, 1999*). We did not detect signals on the timescale of the ultrafast endocytosis observed in *C. elegans* motor neurons and mammalian neurons (*Watanabe et al., 2013a*, *2013b*). If ultrafast endocytosis is present in these sensory neurons, the time constant we measure is likely to represent subsequent reacidification.

The AP180/pCALM homolog UNC-11 is required for efficient vesicle retrieval in AWC$^{ON}$ and ASH. Our results are consistent with current models of AP180/CALM action that emphasize a role in clathrin-dependent sorting at a stage after endocytosis, but before the generation of mature, acidic synaptic vesicles (*Lindner and Ungewickell, 1992*; *Soykan et al., 2017*; *Watanabe et al., 2014*). It is worth noting that this protein has multiple functions in vesicle generation, including effects on the molecular composition and size of synaptic vesicles (*Koo et al., 2015*, *2011*; *Nonet et al., 1999*; *Zhang et al., 1998*).

## Conserved neuronal cell types in divergent animals

How similar, and how divergent, are neuronal cell types in different animals? In genetics, the concept of orthologous genes serves as a valuable framework for cross-species comparisons; whether such orthologous relationships apply to neuronal cells is a subject of debate (*Vergara et al., 2017*). The argument for cell type conservation has been mainly based on developmental transcription factors, rather than mature neuronal properties (*Vergara et al., 2017*). Here, we found that AWC$^{ON}$ neurotransmission, like its sensory signaling, resembles that of vertebrate photoreceptor neurons. Both AWC$^{ON}$ and photoreceptors have basal sensory activity that is suppressed with stimulation, cGMP-based transduction machinery, and circuitry that bifurcates into two streams of ON/OFF neurons (*Chalasani et al., 2007*). We observed that these similarities extend to the synaptic dynamics of AWC$^{ON}$ and zebrafish and goldfish OFF-bipolar neurons, which resemble photoreceptors (*Morgans, 2000*; *Odermatt et al., 2012*). Both AWC$^{ON}$ and OFF-bipolar neurons have fast and slow modes of synaptic vesicle exocytosis and retrieval that are modulated by neuronal activity (*Neves et al., 2001*; *Neves and Lagnado, 1999*). Moreover, synaptic vesicle priming in photoreceptor and bipolar neurons of the mammalian visual system has been suggested to be largely independent of *Munc-13* (*Cooper et al., 2012*), and we observed residual synaptic vesicle release from AWC$^{ON}$ in *unc-13(lf)* mutants. The extensive similarity between vertebrate retinal neurons and *C. elegans* olfactory neurons suggests that they are evolutionarily conserved cell types.

The differences between neuronal cell types across animals are also very substantial – for example, most *C. elegans* neurons do not have sodium-based action potentials (and neither do vertebrate photoreceptors). It remains to be seen how widely the idea of orthologous cell types holds, but it makes specific predictions. For example, we found that ASH and AWC$^{ON}$ sensory neurons had different synaptic dynamics and molecular requirements: are ASH neurons similar to vertebrate nociceptors in their synaptic properties, as they are in their sensory use of TRPV channels? If neurons do fall into conserved classes, the convergence of genetics, behavior, and imaging tools such as pHluorins in *C. elegans* provide an avenue to uncovering their basic properties and underlying molecular mechanisms with single-cell resolution in vivo.

## Materials and methods

### C. elegans culture

C. elegans strains were maintained under standard conditions on NGM plates at 21–22°C and fed OP50 bacteria (Brenner, 1974). Wild-type animals correspond to the Bristol strain N2. Transgenic lines were generated using standard methods by injecting young adult hermaphrodites with the desired transgene and a co-injection plasmid that expresses a fluorescent marker. In some cases, empty vector was included to increase the overall DNA concentration to a maximum of 100 ng/μl. A full strain list and the identity of mutants are presented in Supplementary files 1 and 2.

### Molecular biology

For cell-specific expression in AWC[ON] and ASH we used the promoters str-2 and sra-6, respectively. The VGLUT-pH expression construct was created by subcloning super-ecliptic pHluorin (GenBank AAS66682.1) into the first luminal loop domain of the C. elegans vesicular glutamate transporter eat-4, based on homology to mammalian VGLUT-1 (Voglmaier et al., 2006). Using site-directed mutagenesis (Stratagene quickchange protocol) we inserted a KPN-1 restriction site into the eat-4.a cDNA (wormbase CDS ZK512.6a) after the conserved glycine residue at position 106. Super-ecliptic pHluorin was inserted into this site using primers that added a 14 amino acid linker.

Forward primer:
'5-GAATCGTAGGTACCTCTACCTCTGGAGGATCTGGAGGAACCGGAGG ATCTATGGGAAGTAAAGGAGAAGAACTTT-3

Reverse primer:
'5-GAATCGTAGGTACCTCCGGTTCCTCCAGATCCTCCGGTTCCTCCGG TTCCTCCACCGGTTTTGTATAGTTCATCCA-3'

syGCaMP was created by fusing GCaMP3 to the C-terminus of synaptogyrin-1 (sng-1). sng-1 cDNA (wormbase CDS T08A9.3) was isolated from a N2 whole worm cDNA library and subcloned into a pSM expression vector containing GCaMP3 using AflII and SacII restriction enzymes. sng-1 was fused to GCaMP3 through a 6x Glycine-Serine linker.

Forward primer: 5'- CAAATGATGACAGCGAAGTGGCTTAAGCATGGTATTGATATCTGAGC-3

Reverse primer:5'- GAATCGTAccgcggGAACCACTACCACTACCataaccatatcct tccgactga-3'

To create CD4-pH, Super-ecliptic pHlourin was localized to the extracellular surface by fusion to a modified form of CD4 (Feinberg et al., 2008). CD4-pH was produced by exchanging the spGFP 1–10 in the pSM vector CD4-2::spGFP1-10 (Feinberg et al., 2008) for super-ecliptic pHluorin using the restriction sites Nhe1 and Sal1. The inserted super-ecliptic pHluorin contained the N-terminal linker domain: Gly-Gly–Gly–Gly–Gly-Ser-Gly–Gly–Gly-Ser.

AWC[ON] pkc-1 rescue: pkc-1.a cDNA (corresponding to wormbase's CDS F57F5.5a sequence) was isolated from N2 whole worm cDNA libraries and subcloned into the pSM expression vector using Nhe1 and Kpn1 restriction sites. AWC[ON] expression was achieved using the str-2 promoter (vector str-2:pkc-1.a_cDNA:sl2:mCherry). Expression was confirmed for each animal tested by checking for co-expression of mCherry. pkc-1 cDNA isolation primers:

Forward:
'5-GAATCGTAGCTAGCATGCTGTTCACAGGCACCGTGC-3'

Reverse:
'5-GAATCGTAGGTACCTTAGTAGGTAAAATGCGGATTGA-3'

### Imaging activity-dependent fluorescence reporters

All imaging experiments for a given condition or observation were repeated on at least two separate days using independently prepared buffers and stimuli. The number of trials per animal and the total number of animals are reported along with the total trial number in the figure legends. The interval length for all trials was 30 s in addition to the time recorded before or after the stimulation. For AWC[ON] recordings in which the pulse length was varied, stimulation was ordered sequentially as follows: 6 trials with 10 s pulses, 3 trials with 60 s pulses, and a single 3 min pulse. Order of stimulation did not appear to affect results. Most stimulation protocols involved multiple trials per animal, as detailed in figure legends. Each trial was considered a biological replicate for the purpose of statistical analysis. No more than three trials were conducted per animal except for Figure 2C and 2E (10 s

odor pulses). For all experiments, wild-type controls and mutants were measured in parallel, cycling individual animals from each genotype: wild-type, mutant, repeat.

GCaMP reporters were empirically chosen to match the dynamic range of signaling in the relevant neuron and compartment. The higher-affinity GCaMP5A protein detects both calcium increases and decreases in AWC$^{ON}$ cell body, whereas the lower-affinity GCaMP3 protein preferentially reports the peak calcium levels in AWC$^{ON}$ axons and the ASH cell body, declining only slightly after odor addition to AWC$^{ON}$.

Imaging was conducted on a Zeiss Axiovert 100TV wide-field microscope on animals loaded into custom-built PDMS microfluidic chambers (Chronis et al., 2007). Images of synapses were acquired through a 100 × 1.4 NA Zeiss APOCHROMAT objective onto an Andor ixon +DU-987 EMCCD camera using Metamorph 7.7.6–7.7.8 acquisition software. Camera settings: 14-bit EM-GAIN enabled digitizer (3MHz); baseline clamped; overlapped recording mode; 0.3 uS vertical clock speed; binning = 1. Most experiments used a pre-amplifier gain of 5x. Illumination was provided by a Lumencore SOLA-LE solid-state LED lamp. Illumination input was passed through a 1.3 ND filter. Narrow bandwidth blue light illumination (484–492 nm) was produced using the CHROMA 49904-ET Laser Bandpass filter set. Images were cropped around the head of animal. Stimulus triggering was performed through Metamorph via digital input from a National Instruments NI-DAQmx box to an Automate Valvebank 8 II actuator that triggered Lee Corporation solenoid valves. Custom journals specified pre-programmed recording parameters and performed automated file naming and storage.

TIFF time-stacks were acquired at five frames per second (fps) using a 200 msec acquisition time in most cases; for AWC$^{ON}$ GCaMP5 cell body recordings, TIFF time-stacks were acquired using a 40x objective at 10 fps, 100 msec acquisition time.

Animals were age-synchronized by picking L4s onto fresh NGM OP50 seeded plates 12–18 hr before experiments. For recordings of AWC$^{ON}$ activity, animals were starved for 20–30 min in S basal buffer (Brenner, 1974) prior to loading into the microfluidic chamber. All imaging was conducted in S basal buffer. To prevent movement, animals were paralyzed with 1 mM (-)-Tetramisole hydrochloride (Sigma-Aldrich, St. Louis, MO) during acquisition (Gordus et al., 2015). After loading in the microfluidic chambers, animals were allowed to acclimate for 5 min before imaging. For recordings of ASH activity, a 90 s recording was performed before any stimulation to allow animals to adapt to the blue light.

Butanone (Sigma) or NaCl (Fisher) stimuli were prepared fresh on the day of the experiment from pure stock solutions. The final butanone concentration was 11.2 µM ($10^{-6}$ dilution, prepared by serial $10^{-3}$-fold dilutions) and the final NaCl concentration was 500 mM. Stimulus and control solutions were prepared in S basal buffer in amber glass vials. Control buffer and stimuli were delivered via 30 mL syringe reservoirs (Fisher).

## Acquisition of fluorescence measurements

To extract fluorescence measurements from VGLUT-pHluorin images, we developed custom semi-automated tracking software in ImageJ. Images were first corrected for x-y drift using image registration that placed the axon at a specific set of image coordinates for the entire recording by shifting each frame in x and/or y (Tseng et al., 2011). The microfluidic device prevents most z-plane drift, but images in which significant z-plane drift was detected were discarded. To aid in axon selection, images underwent rolling-ball background subtraction and then were averaged over space (two pixels in x and y) and time (average of time-stack) (Figure 1—figure supplement 2). The entire segment of the axon that was in view was then specified by the user and outlined by hand with the aid of pixel intensity thresholding. From this axon outline, intensity and pixel information was extracted from the raw drift-corrected recording along with local background measurements along the axon (Figure 1—figure supplement 2E). This process was also used to acquire measurements of AWC$^{ON}$ syGCaMP images.

The sra-6p promoter used for ASH imaging is also expressed in the ASI and PVQ neurons, whose axons are posterior to the ASH axon. VGLUT-pH fluorescence could also be detected in these other axons (mainly PVQ) but did not significantly respond to stimulation. In any given experiment, we recorded VGLUT-pH signals from a single ASH axon, either the on right or left side of the animal. Because of ASI and PVQ VGLUT-pH fluorescence, only the anterior background ROIs were used in analysis of ASH VGLUT-pH recordings.

For cell body measurements of GCaMP responses in AWC[ON] and ASH we used a custom written ImageJ script (*Gordus et al., 2015*) to track cell body position and extract intensity measurements.

To validate the analysis of the entire axon as a single integrated measurement in VGLUT-pH experiments, we measured VGLUT-pH responses from small equally spaced ROIs along the axon. These were obtained by taking the outlined axon segment as in (*Figure 1—figure supplement 2E*) and cutting the axon into smaller ROI segments, each eight pixels long in the y-axis, and performing a correlation analysis on a dataset of AWC[ON] VGLUT-pH recordings from wild-type animals stimulated with a 60 s pulse of 11.2 uM butanone. For 120 stimulated axons, all ROIs along the axon were positively correlated with the mean integrated axon fluorescence (*Figure 1—figure supplement 2F*). There was scatter in the strength of this correlation across different ROIs, potentially consistent with heterogeneity among synaptic regions (*Figure 1—figure supplement 2F*), but given the small magnitude of the signal this possibility was not examined further.

For technical reasons, we were unable to estimate the total releasable vesicle pool by the methods used in other systems. Attempts at fluorescence dequenching by soaking the animal in $NH_4Cl$ were unsuccessful. Strong stimuli, which are used to deplete the vesicle pool in other systems, inhibit AWC[ON] rather than activating it, and ASH habituates rapidly to stimulation.

## Quantification of fluorescence changes; background and bleaching correction

In all traces, dF/F was calculated as:

$$\frac{dF}{F_o}(t) = \left( \frac{F(t) - F_o}{F_o} \right)$$

(1)

$F(t)$=fluorescence of the ROI minus the corrected background at time (t)

$F_o$ = fluorescence of the ROI minus the background at a reference time point or value.

For AWC[ON], $F_o$ was set to the mean background-corrected signal two seconds before odor removal. For ASH recordings, $F_o$ was set to the mean background-corrected signal two seconds before the first stimulus for each trial. In each case, $F_o$ corresponded to a stable baseline within and across recordings.

VGLUT-pH has a low baseline signal that is relatively close in intensity to the background autofluorescence generated by the surrounding tissue of the head. This became problematic when attempting to apply background subtraction and perform comparisons of fluorescence change using the deltaF function (dF/F) described above (*Equation 1*). The deltaF function normalizes the fluorescence change to its baseline intensity, allowing a comparison of signal changes between conditions with different absolute values, but is highly sensitive to fluctuations in $F_o$ when $F_o$ is small. For VGLUT-pH, a further confound is created by reporter localization to parts of secretory pathway(s) that do not participate in synaptic release. These can be seen as small puncta that contribute to $F_o$ but not to fluorescence changes. To prevent background correction from shifting $F_o$ into a realm where small variations in $F_o$ generate large fluctuations in deltaF, we subtract only the fluctuations in the background signal (*Equation 2*):

$$BG_{Delta} = BG(t) - BGmin$$

(2)

$BG$=background at time (t)

$BGmin$=minimum background value during the recording.

VGLUT-pH has very slow bleaching kinetics (*Ariel and Ryan, 2010*; *Balaji and Ryan, 2007*), essentially showing little to no bleaching over the timecourse of our recordings. This is likely the result of the reporter existing mainly within synaptic vesicles in the quenched state, and the rapid cycling of the fluorescent form back into this state. Unlike the VGLUT-pH reporter, background autofluorescence does show significant bleaching, and therefore using this background signal for correction can result in an artificial increase in the VGLUT-pHuorin signal over time. To avoid this artifact, we corrected for background bleaching before performing background correction. Bleaching was assumed to be approximately linear and was modeled by line fitting using the Matlab function 'polyfit.' A threshold for specifying significant bleaching was set to a 0.5% drop in mean fluorescence intensity over 2 min of recording time. Bleaching was corrected by subtracting the linear fit from the background signal.

## Statistical analysis and curve fitting

All statistical tests are indicated in figure legends, and were performed using Prism 7 GraphPad software. Additional supporting statistical analysis for each figure can be found in the corresponding Source Data files. Unless otherwise stated, all acquired data for a given condition were included in the analysis. A given genotype or condition was tested in at least four animals recorded over at least two different days, unless the same condition was replicated in another presented dataset. This minimum number was based on the high reliability of stimulus-evoked responses in wild-type animals (non-responders/trials: ASH, 5/204; AWC$^{ON}$, 2/132). For mutant strains, data were collected for at least ten trials and typically for 20–30 to enable detection of intermediate effects.

Fitting of decay curves was performed in Matlab using the 'fit' function with a custom 'fitType' equation. For single exponential fits fitType = a1*exp (-x/tau1)+C. For double exponential fits fitType = a1*exp (-x/tau1)+a2*exp (-x/tau2)+C. For triple exponential fits fitType = a1*exp (-x/tau1)+a2*exp (-x/tau2)+a3*exp (-x/tau3)+C. In each case, all 'a' and 'tau' variables are fitted parameters and are constrained to be greater than zero. C = the baseline to which traces decayed, normalized to zero. Unless stated otherwise, fitting was applied to the entire decay period of the trace, which corresponds to the odor-addition phase for AWC$^{ON}$ or the 20 s immediately following NaCl removal for ASH. For comparison of AWC$^{ON}$ decays during different stimulus durations (60 s vs 20 s, *Figure 7*), we fit both traces using the initial 20 s to keep comparisons consistent. Decay constant averages reported for a given condition were performed as follows: each individual trace for a given data set was fitted individually, and each fit was plotted on its trace and then inspected. Traces that could not be fit due to high noise or did not exhibit a decay were removed from the analysis and is reflected in the change in n reported in the figure legends.

## Comparing exponential fits using AIC$_C$

To compare models we used the Akaike's Information Criterion (AIC) as described in (*Motulsky and Christopoulos, 2004*): $AIC = N * \ln\left(\frac{SS}{N}\right) + 2K$, where N = number of data points, K = number of parameters, and SS = is the sum of the square of the vertical distances of the points from the curve. We used the corrected version for small N: AIC$_C$ = $AIC + \frac{2K(K+1)}{(N-K-1)}$. AIC$_C$ was performed for the fit on each individual trace. To determine of the relative likelihood of two models, we computed the probability that one model is more likely than the other (AIC$_P$): AIC$_P$ = $\frac{e^{-0.5\Delta}}{1 + e^{-0.5\Delta}}$, where $\Delta$ = the difference in AIC$_C$ scores. All AIC-based analysis was conducted in Matlab using custom scripts.

## Butanone chemotaxis assays

Butanone chemotaxis assays were conducted on square plates containing 20 ml chemotaxis agar (1.6% Agar, 5 mM potassium phosphate buffer pH 6.0, 1 mM CaCl2, 1 mM MgSO4) poured 18–24 hr before the assay. Adult animals were removed from NGM growth plates with chemotaxis buffer (5 mM potassium phosphate buffer pH 6.0, 1 mM CaCl2, 1 mM MgSO4), transferred to 1.5 ml microcentrifuge tubes (Eppendorf) and washed twice with chemotaxis buffer. 1 µl of 1 M sodium azide spotted at the location of odor and control spots was used to immobilize animals that reach odor sources. ~100–350 animals were spotted onto the center of the chemotaxis plate. Two 1 µl spots of butanone diluted in ethanol (1:1000) were spaced on one edge of the plate, opposite to two 1 µl spots of ethanol control. The liquid drop containing the animals was then wicked away to start the assay. Assays were allowed to run for 1–2 hr and then moved to 4°C prior to counting. The chemotaxis index was calculated as = (# animals on odor side – # animals on control side) / (Total # counted animals).

## Acknowledgements

We thank Aditya Rangan for guidance in kinetic modeling, Andrew Gordus for sharing his image analysis code, Tim Ryan and Jeremy Dittman for extensive discussions of synaptic imaging, and Johannes Larsch, Aylesse Sordillo, Qiang Liu, Sagi Levy, and Daniel Colon-Ramos for discussions and comments on the manuscript. This work was supported by the Howard Hughes Medical Institute and by a gift from the Jensam Foundation.

## Additional information

### Funding

| Funder | Author |
| --- | --- |
| Howard Hughes Medical Institute | Cornelia I Bargmann |
| Jensam Foundation | Cornelia I Bargmann |

The funders had no role in study design, data collection and interpretation, or the decision to submit the work for publication.

### Author contributions

Donovan Ventimiglia, Conceptualization, Data curation, Software, Formal analysis, Investigation, Writing—original draft, Writing—review and editing; Cornelia I Bargmann, Conceptualization, Supervision, Funding acquisition, Writing—original draft, Writing—review and editing

### Author ORCIDs

Cornelia I Bargmann  http://orcid.org/0000-0002-8484-0618

### Decision letter and Author response

Decision letter https://doi.org/10.7554/eLife.31234.029
Author response https://doi.org/10.7554/eLife.31234.030

## Additional files

### Supplementary files

• Source code 1. Tracking and analysis source codes
DOI: https://doi.org/10.7554/eLife.31234.024

• Supplementary file 1. Strain list
DOI: https://doi.org/10.7554/eLife.31234.025

• Supplementary file 2. Molecular identities of mutations
DOI: https://doi.org/10.7554/eLife.31234.026

• Supplementary file 3. (A) Supporting statistics tables for *Figure 3* (B) Supporting statistics tables for *Figure 4* (C) Supporting statistics tables for *Figure 8*.
DOI: https://doi.org/10.7554/eLife.31234.027

• Transparent reporting form
DOI: https://doi.org/10.7554/eLife.31234.028

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
