## [Decision Letter]

Thank you for submitting your article "Diverse modes of synaptic signaling, regulation, and plasticity distinguish classes of *C. elegans* glutamatergic neurons" for consideration by *eLife*. Your article has been favorably evaluated by a Senior Editor, Graeme Davis as the Reviewing Editor, and three reviewers. The following individual involved in review of your submission has agreed to reveal his identity: Joshua M Kaplan (Reviewer #3).

Summary of reviewer discussion:

The authors describe an elegant analysis of neurotransmitter release at two central synapses in the *C. elegans* brain. This is the very first such analysis to have been described at any *C. elegans* central synapse and the authors describe new optical probes that were necessary for such an analysis (particularly the first use of VGLUT/pHluorin in *C. elegans*). Thus, this work represents an obvious technical advance for *C. elegans* neurobiology. Beyond their technical achievements, the authors also discover some intriguing differences between the two sensory synapses they study (AWC and ASH). Mutations in core presynaptic machinery produce very different effects at these two synapses (including critical SNARE binding proteins), highlighting the fact that different synapses operate via very different molecular mechanisms. The work has been performed to a high standard, and the text is very clearly written, making all the important points without undue speculation. All of the referees were in agreement with respect to the suitability of the work for publication in *eLife*. However, the reviewers have some suggestions that the authors may want to consider prior to resubmission of the final work. Please note that, given the high level of enthusiasm, it is expected that the turnaround upon the next submission should be rapid.

Minor Comments:

1) The authors find that two different *pkc-1* alleles (*nj1* and *nj3*) have very different effects on AWC synapses (Figure 8—figure supplement 1). The missense allele (*nj1*) has a much stronger defect than the early nonsense allele (*nj3*). I am concerned that the *nj1* allele has a dominant negative effect, which may produce more severe phenotypes than null alleles. This concern is somewhat offset by the fact that the *nj1* synaptic exocytosis defects are rescued by a transgene containing a wild type copy of *pkc-1*. Differences in the impact of the *pkc-1(nj1)* mutation on AWC and ASH is one of the main findings of the paper. If *nj1* has a non-null phenotype, that changes how I would interpret this difference (e.g. maybe on and off rates of PKC-1 are different at AWC and ASH, and this results in differences in the severity of the *nj1* dominant negative phenotype). This concern would be eliminated if the authors show that nj1 is similar to a true *pkc-1* null (produced by CRISPR gene editing). I realize that this would require some effort. Alternatively, the authors could modify the text to indicate that alternative interpretations of the *nj1* phenotypes are still possible.

2) The VGLUT/pH signals are fairly small (~4% change in total puncta fluorescence). In other preps (e.g. cultured mammalian neurons), these signals are normalized to the total releasable pool (either by dequenching with NH4Cl, or by comparing to the maximal signal produced by high frequency stimulation). It would be reassuring if the worm VGLUT/pH responses represent a similar fraction of the total releasable pool to that observed in mammalian neurons. Perhaps the authors already have such data?

3) As seen in mammalian cells, the rise time of the VGLUT/pH responses is fairly slow (seconds) compared to the extremely fast kinetics of CaV activation and SV fusion measured electrophysiologically (which occur in milliseconds). This most likely reflects the slow kinetics of odorant exchange and chromophore dequenching. Nonetheless, this raises the concern that the VGLUT/pH signals measured do not correspond to sensory evoked SV fusions (but instead some slower form of release). It would be reassuring to show that the VGLUT/pH signals are coupled to the same CaV channels that drive SV fusion (e.g. UNC-2/CaV2 channels), and exhibit a similar tight coupling to CaV channels (as evidenced by resistance to EGTA-AM inhibition).

4) One of the interesting findings reported here is that *unc-13* mutations eliminate ASH responses but not AWC responses whereas the converse effects are observed in *unc-18* mutants. Are the residual responses in *unc-13* (in AWC) and *unc-18* (in ASH) significant? It is possible that some of these differences are caused by non-autonomous circuit effects. For example, the relatively larger residual AWC synaptic response in *unc-13* mutants could reflect decreased inhibition of AWC by other neurons. This could be addressed by showing cell autonomous rescue of the *unc-13* and *unc-18* defects.

5) Do *unc-31*/CAPS mutations have different effects on AWC and ASH synaptic responses? This could explain why the *unc-13* results differ for these two neurons.

6) Another interesting result is that sensory evoked responses are exaggerated in *cpx-1* mutants. I am not aware of similar findings in other preps. Typically complexin mutants have smaller evoked responses, which makes this result pretty interesting. Unless I missed other published precedents, the authors may want to comment on the novelty of this result and its implications for complexin synaptic function in intact animals.

7) Is the amplitude of AWC GCAMP responses increased in *pkc-1(nj1)* mutants? The data shown in Figure 8 indicates a trend in that direction.

---

## [Author Response]

1) The authors find that two different pkc-1 alleles (nj1 and nj3) have very different effects on AWC synapses (Figure 8—figure supplement 1). The missense allele (nj1) has a much stronger defect than the early nonsense allele (nj3). I am concerned that the nj1 allele has a dominant negative effect, which may produce more severe phenotypes than null alleles. This concern is somewhat offset by the fact that the nj1 synaptic exocytosis defects are rescued by a transgene containing a wild type copy of pkc-1. Differences in the impact of the pkc-1(nj1) mutation on AWC and ASH is one of the main findings of the paper. If nj1 has a non-null phenotype, that changes how I would interpret this difference (e.g. maybe on and off rates of PKC-1 are different at AWC and ASH, and this results in differences in the severity of the nj1 dominant negative phenotype). This concern would be eliminated if the authors show that nj1 is similar to a true pkc-1 null (produced by CRISPR gene editing). I realize that this would require some effort. Alternatively, the authors could modify the text to indicate that alternative interpretations of the nj1 phenotypes are still possible.

We have added data for two additional *pkc-1* alleles, *pkc-1(nu448)* and *pkc-1(ok563)* to Figure 8—figure supplement 1. *pkc-1(nu448)* is a nonsense allele that terminates the protein within the kinase domain in all isoforms. *pkc-1(ok563)* is a 1.3kb deletion that removes the conserved C2 domain in two of the three *pkc-1* isoforms.

All four alleles strongly decrease the response to odor addition in AWC^ON^, and all diminish but do not eliminate evoked release after odor removal. The two alleles that affect the kinase domain (point mutation *nj1*, termination *nu448*) have stronger phenotypes than the two alleles that affect N-terminal regions (termination *nj3,* deletion *ok563).* Heterogeneity among *pkc-1* alleles has also been observed in behavioral assays (Hyde et al., 2011).

As indicated by the reviewer, the ability to rescue *pkc-1(nj1)* with a wild-type transgene argues against the allele representing a gain of function. There are four other PKCs encoded in the *C. elegans* genome, with *tpa-1* having a redundant role in the response to the DAG agonist PMA (Okochi et al., 2005). By analogy with other kinase pathways (Madhani et al., 1997), we speculate that the stronger phenotype may reflect compensation between PKC-1 and other PKCs when no PKC-1 protein is present. The *nj1* point mutation, which likely preserves PKC-1 kinase-independent interactions, might block such compensatory actions. We have added this speculation to the Discussion.

2) The VGLUT/pH signals are fairly small (~4% change in total puncta fluorescence). In other preps (e.g. cultured mammalian neurons), these signals are normalized to the total releasable pool (either by dequenching with NH4Cl, or by comparing to the maximal signal produced by high frequency stimulation). It would be reassuring if the worm VGLUT/pH responses represent a similar fraction of the total releasable pool to that observed in mammalian neurons. Perhaps the authors already have such data?

We do not have any data that would lend insight into the total releasable pool. Attempts at dequenching with NH_4_Cl by soaking the animal were unsuccessful, and attempts at strong stimulation to exhaust the releasable pool were problematic due to basal activity (AWC) or rapid adaptation (ASH). We added this caveat to the Materials and methods section. To address questions regarding the releasable SV pool, we recommend future studies to combine pHluorin imaging with either dequenching or electrophysiology in dissected preparations.

3) As seen in mammalian cells, the rise time of the VGLUT/pH responses is fairly slow (seconds) compared to the extremely fast kinetics of CaV activation and SV fusion measured electrophysiologically (which occur in milliseconds). This most likely reflects the slow kinetics of odorant exchange and chromophore dequenching. Nonetheless, this raises the concern that the VGLUT/pH signals measured do not correspond to sensory evoked SV fusions (but instead some slower form of release). It would be reassuring to show that the VGLUT/pH signals are coupled to the same CaV channels that drive SV fusion (e.g. UNC-2/CaV2 channels), and exhibit a similar tight coupling to CaV channels (as evidenced by resistance to EGTA-AM inhibition).

There are two issues here: the long-duration sensory stimuli that we use in the in vivoexperiments (10-60s), compared to the short stimuli used to evoke synaptic vesicle fusion ex vivo, and the nature of the VGLUT reporter. With respect to the stimulus, the onset of the VGLUT signal (change in slope) is as fast as the onset of the calcium signal and as fast as we can detect with our recording system (e.g. Figure 1, Figure 2). Based on both calcium signals and behavioral output, the neurons remain active for some time after stimulus onset (ASH) or removal (AWC), so VGLUT should continue to be mobilized.

With respect to the reporter, the slow rise and decay times with pHluorins are reflective of the entire cycle of vesicle release, endocytosis, and re-acidification (with the last being slow). The dynamics of calcium and pHlourin rise and decay fall out of known properties of synapses – calcium buffering/extrusion is faster than vesicle reacidification, and therefore the pHluorin signal is effectively cumulative compared to the calcium signal. Our results closely resemble those in mammalian ex vivo preparations, where the pHluorin signal is also essentially the integral of the calcium signal. See, for example, the instantaneous rise in pHlourin signal to a single action potential in Figure 1 of Ariel and Ryan, 2010, compared to the slow cumulative rise over 90s during a train of action potentials in the same figure. The results in our Figure 2 quantitatively support this interpretation for *C. elegans*, showing that the derivative of the VGLUT signal resembles the calcium signal, with both signals starting immediately and then sharing an inflection point 0.7s after stimulus onset.

The signals observed here track with well-defined SNARE mutations, endocytosis mutants, and tetanus toxin in ways that strongly support their association with synaptic vesicle release. We make this point in the revised Discussion. Whether there are multiple classes of synaptic vesicle release in vivois an interesting question for future study. Certainly at the *C. elegans* NMJ, basal and evoked release appear to be somewhat different.

We did not monitor calcium sources in this paper, but previous work indicates that *unc-2* (CaV2) is not the only source of synaptic calcium in AWC^ON^ (Saheki and Bargmann, 2009), and ASH signaling involves *unc-2* CaV2 channels, *egl-19* CaV1 channels, and calcium release from intracellular stores (Zahratka et al., 2015). Similarly, both CaV2 and CaV1 contribute to synaptic transmission and specifically tonic release at the *C. elegans* NMJ, as demonstrated in a recent paper that we now cite in the Discussion (Tong et al., 2017). We state in the Discussion that further study of calcium sources should be conducted. Again, this will require a dissected preparation – intact *C. elegans* is not affected by EGTA.

4) One of the interesting findings reported here is that unc-13 mutations eliminate ASH responses but not AWC responses whereas the converse effects are observed in unc-18 mutants. Are the residual responses in unc-13 (in AWC) and unc-18 (in ASH) significant? It is possible that some of these differences are caused by non-autonomous circuit effects. For example, the relatively larger residual AWC synaptic response in unc-13 mutants could reflect decreased inhibition of AWC by other neurons. This could be addressed by showing cell autonomous rescue of the unc-13 and unc-18 defects.

The *unc-13* and *unc-18* residual responses are significant and different from each other; we now make this point and refer to the supplementary statistical table in the text. We agree there is potential for circuit effects in all of the mutants other than *pkc-1,* for which we demonstrate cell-specific rescue, and we state this point clearly in the Discussion.

5) Do unc-31/CAPS mutations have different effects on AWC and ASH synaptic responses? This could explain why the unc-13 results differ for these two neurons.

We did not examine *unc-31* in this study.

6) Another interesting result is that sensory evoked responses are exaggerated in cpx-1 mutants. I am not aware of similar findings in other preps. Typically complexin mutants have smaller evoked responses, which makes this result pretty interesting. Unless I missed other published precedents, the authors may want to comment on the novelty of this result and its implications for complexin synaptic function in intact animals.

We agree that this a potentially interesting finding, and we have added a statement to this effect to the Discussion. However, this is another case for which we have not demonstrated cell-specific rescue, and we would prefer to be conservative in interpreting it.

7) Is the amplitude of AWC GCAMP responses increased in pkc-1(nj1) mutants? The data shown in Figure 8 indicates a trend in that direction.

Yes, there is a small but significant difference in the peak magnitude of the responses (mean difference of 13.64 ± 4.473%). We are reluctant to put too much emphasis on this small effect, but we now mention it in the text and present the analysis in Figure 8—source data 1 under supporting statistics. The small increase in calcium response contrasts with the opposite effect that *pkc-1(lf)* has on AWC^ON^ synaptic release.